# Hsp90-mediated regulation of DYRK3 couples stress granule disassembly and growth via mTORC1 signaling

Laura Mediani[1], Francesco Antoniani[1,†], Veronica Galli[1,†], Jonathan Vinet[1,2], Arianna Dorotea Carrà[1], Ilaria Bigi[1], Vadreenath Tripathy[3], Tatiana Tiago[1], Marco Cimino[1], Giuseppina Leo[1], Triana Amen[4] (iD), Daniel Kaganovich[4] (iD), Cristina Cereda[2] (iD), Orietta Pansarasa[2], Jessica Mandrioli[5], Priyanka Tripathi[6], Dirk Troost[7], Eleonora Aronica[7], Johannes Buchner[8] (iD), Anand Goswami[6], Jared Sterneckert[3], Simon Alberti[9,*] (iD) & Serena Carra[1,**] (iD)

## Abstract

**Stress granules (SGs) are dynamic condensates associated with protein misfolding diseases. They sequester stalled mRNAs and signaling factors, such as the mTORC1 subunit raptor, suggesting that SGs coordinate cell growth during and after stress. However, the molecular mechanisms linking SG dynamics and signaling remain undefined. We report that the chaperone Hsp90 is required for SG dissolution. Hsp90 binds and stabilizes the dual-specificity tyrosine-phosphorylation-regulated kinase 3 (DYRK3) in the cytosol. Upon Hsp90 inhibition, DYRK3 dissociates from Hsp90 and becomes inactive. Inactive DYRK3 is subjected to two different fates: it either partitions into SGs, where it is protected from irreversible aggregation, or it is degraded. In the presence of Hsp90, DYRK3 is active and promotes SG disassembly, restoring mTORC1 signaling and translation. Thus, Hsp90 links stress adaptation and cell growth by regulating the activity of a key kinase involved in condensate disassembly and translation restoration.**

**Keywords** DYRK3; FUS-ALS; Hsp90; phase separation; stress granules
**Subject Category** Translation & Protein Quality

## Introduction

Stress granules (SGs) are dynamic ribonucleoprotein assemblies that form in response to stress conditions and transiently sequester translationally stalled mRNAs along with 40S ribosomal subunits, enzymes, and signaling molecules, such as the p90 ribosomal S6 kinase and the mTORC1 subunit raptor (Kedersha *et al*, 2013). Raptor-mediated sequestration of mTORC1 inside SGs protects cells from apoptosis (Thedieck *et al*, 2013). Upon stress relief, the dual-specificity tyrosine-phosphorylation-regulated kinase 3 (DYRK3) promotes SG dissolution, with subsequent release and reactivation of the mTORC1 complex (Wippich *et al*, 2013). This suggests that SGs are signaling hubs that coordinate cell growth and metabolism in response to changes in environmental conditions, thus enabling the adaptation of cells to stressors (Kedersha *et al*, 2013). However, the molecular details of how cellular signaling regulates SG assembly and dissolution have so far eluded us.

Recent work suggests that impaired SG dynamics are associated with cell dysfunction and death. SG dysfunctionality is emerging as an important pathomechanism in a number of age-related neurodegenerative diseases including amyotrophic lateral sclerosis (ALS), frontotemporal lobar degeneration (FTD), and Alzheimer disease (AD). In these diseases, failure to disassemble SGs has been linked to disturbances in RNA metabolism and to the development of aggregates that cause neuronal death (Zhang *et al*, 2019). Hence, identifying and characterizing the factors that govern SG dynamics

1  Department of Biomedical, Metabolic and Neural Sciences, Centre for Neuroscience and Nanotechnology, University of Modena and Reggio Emilia, Modena, Italy
2  Genomic and Post-Genomic Center, IRCCS Mondino Foundation, Pavia, Italy
3  Center for Regenerative Therapies TU Dresden, Technische Universität Dresden, Dresden, Germany
4  Department of Experimental Neurodegeneration, University Medical Center Göttingen, Göttingen, Germany
5  Department of Neuroscience, St. Agostino Estense Hospital, Azienda Ospedaliero Universitaria di Modena, Modena, Italy
6  Institute of Neuropathology, RWTH Aachen University Hospital, Aachen, Germany
7  Department of (Neuro)Pathology, Amsterdam Neuroscience, Amsterdam UMC, University of Amsterdam, Amsterdam, The Netherlands
8  Center for Integrated Protein Science Munich at the Department Chemie, Technische Universität München, Garching, Germany
9  Biotechnology Center (BIOTEC), Center for Molecular and Cellular Bioengineering (CMCB), Technische Universität Dresden, Dresden, Germany
   *Corresponding author. Tel: +49 351 46340243; E-mail: simon.alberti@tu-dresden.de
   **Corresponding author (lead contact). Tel: +39 059 2055265; E-mail: serena.carra@unimore.it
   †These authors are contributed equally to this work as second, third authors

will provide targets with therapeutic potential to treat these uncurable diseases. Surprisingly, although we gained detailed knowledge on the mechanisms that promote SG assembly (Kedersha *et al,* 1999; Matsuki *et al,* 2013; Van Treeck *et al,* 2018), the mechanisms regulating SG disassembly are still poorly understood.

RNA-binding proteins (RBPs) along with high concentrations of free RNA are the crucial factors that drive SG assembly (Bounedjah *et al,* 2012; Ciryam *et al,* 2015; Kroschwald *et al,* 2015; Van Treeck *et al,* 2018; Guillen-Boixet *et al,* 2020; Sanders *et al,* 2020; Yang *et al,* 2020). However, many of the RBPs that assemble into SGs have a high propensity to misfold and aggregate, which can affect the material properties of SGs and their ability to dissolve (Molliex *et al,* 2015; Patel *et al,* 2015). Such aberrant SGs can also arise from the accumulation of misfolding-prone proteins, such as defective ribosomal products (DRiPs) or ALS-linked C9orf72 dipeptide repeat proteins, inside SGs (Ganassi *et al,* 2016; Lee *et al,* 2016; Mateju *et al,* 2017; Guillen-Boixet *et al,* 2020). As a result, SGs are under close surveillance by the protein quality control (PQC) system. The PQC system includes molecular chaperones, which recognize aberrantly folded proteins and refold misfolded proteins, as well as the ubiquitin–proteasome and autophagy degradation systems, which clear misfolded proteins (Balchin *et al,* 2016). Indeed, molecular chaperones such as Hsp70 and valosin-containing protein (VCP) are able to prevent the accumulation of misfolded proteins inside SGs, maintaining SG dynamics (Ganassi *et al,* 2016; Mateju *et al,* 2017). In addition, molecular chaperones are required to disassemble aberrant SGs. When surveillance by the PQC machinery fails, persisting aberrant SGs are targeted by autophagy or proteasomal degradation involving the factors p62/SQSTM1 and ZFAND1/VCP (Buchan *et al,* 2013; Mateju *et al,* 2017; Chitiprolu *et al,* 2018; Turakhiya *et al,* 2018). However, our understanding of SG surveillance is still in its infancy and we do not know whether other factors are required to maintain SGs in a healthy state.

One essential and ubiquitous molecular chaperone required for the folding and maturation of a large variety of proteins, including SG components, is Hsp90 (Jain *et al,* 2016; Markmiller *et al,* 2018). Hsp90 has been suggested to regulate the assembly of processing bodies (PBs) (Matsumoto *et al,* 2011), cytosolic RNA-protein condensates involved in mRNA storage (Standart & Weil, 2018). Moreover, Hsp90 promotes, through a yet unknown mechanism, the recruitment of argonaute 2, eIF4E, and its binding partner eIF4E transporter (4E-T) into SGs (Pare *et al,* 2009; Suzuki *et al,* 2009). In agreement with this, a recent proteomic analysis in a fungal pathogen identified a novel role for Hsp90 in stabilizing PB and SG components (O'Meara *et al,* 2019). However, despite the well-known function of Hsp90 in stress tolerance, it is unclear whether Hsp90 has a role in regulating SG dynamics or functionality.

Here, we use genetics and fluorescence microscopy of fixed and live cells to study the functional effects of Hsp90 on SGs in mammalian cells.

## Results

### Inhibition of Hsp90 delays stress granule disassembly in an Hsp70-independent manner

In mammalian cells, the majority of SGs disassemble in the stress recovery phase, and only a minor fraction of SGs (< 5–10%) is targeted to autophagy for clearance (Ganassi *et al,* 2016). Using time-lapse microscopy and HeLa-Kyoto cells expressing the SG marker G3BP2-GFP, we confirmed that most sodium arsenite induced SGs disassemble when the stress is removed; SGs dissolved regardless of whether lysosomal proteases were inactivated with ammonium chloride (Fig EV1A and Movie EV1), suggesting that autophagy does not play a major role. By contrast, inhibition of the ATPase activity of Hsp70 with VER-155008 resulted in SG persistence (VER; Figs 1A and EV1B, and Movie EV1), in agreement with the literature (Ganassi *et al,* 2016; Mateju *et al,* 2017). However, SG persistence was not absolute: SGs could still disassemble in the presence of the Hsp70 inhibitor, although with slower kinetics and SGs persisted only in ca. 20–30% of the cells after 4 h of arsenite removal. Combined these data suggest that SG disassembly is regulated by additional mechanisms besides autophagy and Hsp70.

Hsp90 is an essential chaperone (Taipale *et al,* 2010; Biebl & Buchner, 2019; Moran Luengo *et al,* 2019) that interacts with SG components (Markmiller *et al,* 2018). Yet, it is currently unknown whether Hsp90 regulates SG dynamics. To test this idea, we induced SGs with arsenite and we measured their disassembly kinetics in the presence of two well-established inhibitors of the Hsp90 ATPase activity, geldanamycin (GA) and 17-Allylamino-17-demethoxygeldanamycin (17AAG) (Schulte & Neckers, 1998). As a control experiment, we determined whether Hsp90 inhibition induces spontaneous SG assembly. GA or 17AAG treatment alone for up to 4 h induced SGs in only 3% of the cells (Fig EV1B). By contrast, when we added GA or 17AAG during the recovery phase after sodium arsenite treatment, both inhibitors delayed SG disassembly (Fig 1A and B and Movie EV2), with 17AAG showing a dose-dependent effect (Fig 1B and Movie EV2). Delayed SG disassembly upon Hsp90 inhibition could be reproduced in a different cell line (Fig EV1C and D, and Movies EV3 and EV4, PABPC1-Dendra2 HEK293T cells) and was also observed when Hsp90 levels were reduced by siRNA-mediated knockdown (Fig 1C and D and Movie EV5). Of note, Hsp90 inhibition also impaired the disassembly of SGs induced by other types of stress, such as heat shock and treatment with the proteasome inhibitor MG132 (Fig 1E and F). Thus, Hsp90 is a regulator of SG disassembly, independent of stress type and cell line.

Delayed SGs disassembly (or SG persistence) has been observed when misfolding-prone proteins such as DRiPs accumulate in SGs, for example, in cells where Hsp70 is dysfunctional (Ganassi *et al,* 2016). We thus asked whether Hsp90 inhibition affects the accumulation of DRiPs inside SGs. Although Hsp90 has an important role in assisting the folding of a subset of newly synthesized proteins (Schopf *et al,* 2017), we did not observe colocalization of Hsp90 alpha and beta with DRiPs (Fig EV1E). This is in contrast to HSPA8 and HSPA1A (Fig EV1E), two Hsp70 proteins with a well-established role in targeting DRiPs for clearance (Hartl & Hayer-Hartl, 2002; Ganassi *et al,* 2016).

Then, we induced SGs by heat shock in the absence or presence of the chaperone inhibitors VER, 17AAG, or GA, and we quantified DRiP enrichment inside SGs by microscopy. In contrast to Hsp70 inhibition, which caused a strong enrichment of DRiPs inside SGs (Fig EV1F; Ganassi *et al,* 2016), Hsp90 inhibition led to a milder

accumulation of DRiPs (Fig EV1F). Next, we asked if upregulation of Hsp70, which assists DRiP degradation and prevents DRiPs accumulation inside SGs (Ganassi *et al*, 2016), could rescue SG dissolution in cells treated with the Hsp90 inhibitors. However, we found that induction of Hsp70 (Fig EV1G) could not restore SG disassembly in cells treated with 17AAG or GA (Fig EV1H). Together, these data demonstrate that Hsp90 regulates SG disassembly in a different manner than Hsp70.

## Short-term inhibition of Hsp90 does not affect the dynamics of P-bodies

It has been shown that long-term treatment with GA decreases the PB number (Suzuki *et al*, 2009). PBs are cytoplasmic mRNP granules that are present in growing cells and increase in size and number upon stress conditions (Kedersha *et al*, 2005). PBs are intimately connected with SGs: both are in dynamic equilibrium, share

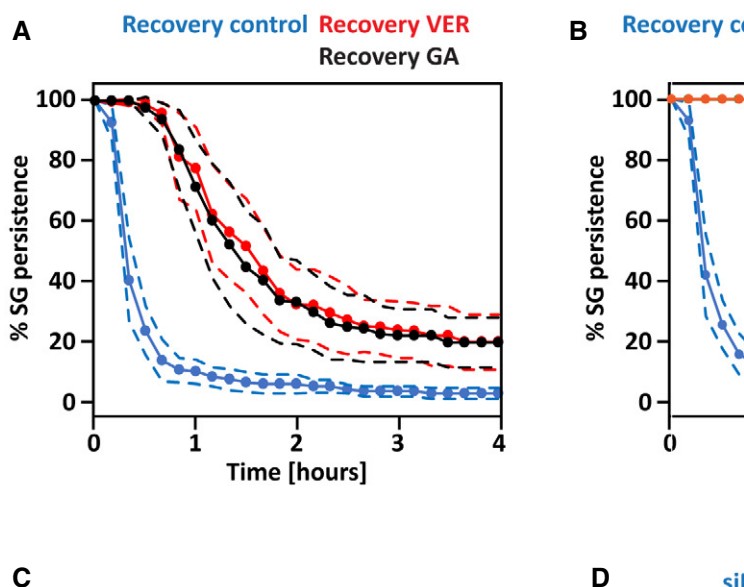

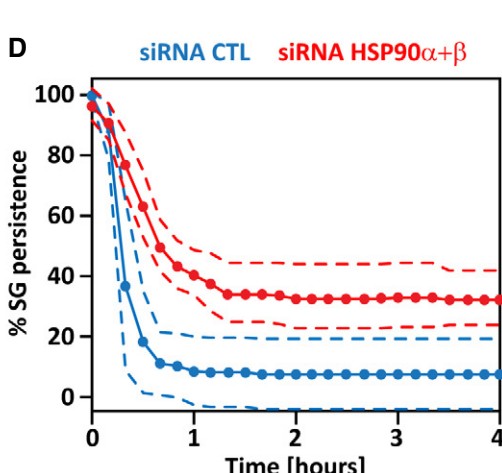

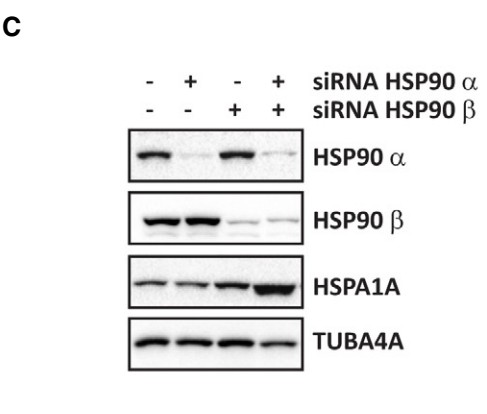

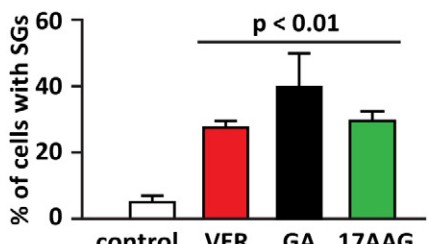

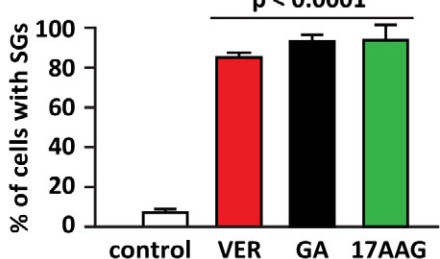

**Figure 1.**

**Figure 1. Inhibition of Hsp90 delays the dissolution of SGs induced by different types of stressors.**

A, B Kinetics of SG dissolution of living cells in absence and presence of Hsp90 or Hsp70 inhibitors. G3BP2-GFP HeLa-Kyoto cells were treated with sodium arsenite (50 μM) for 45 min. Then, cells were allowed to recover in drug-free medium (Recovery Control) or in presence of GA (5 μM), 17AAG (5 μM or 0.5 μM) or VER (40 μM). Images were taken over a time period of 4 h every 10 min. Dashed lines = 95% confidence intervals. Number of cells counted: 193 (Recovery Control, A); 340 (GA 5 μM); 187 (17AAG 5 μM); 209 (VER 40 μM); 363 (Recovery Control, B); 125 (17AAG 0.5 μM); and 353 (17AAG 1.5 μM).

C, D G3BP1-mCherry HeLa-Kyoto cells were lipofected for 72 h with a non-targeting siRNA control or siRNAs specific for Hsp90 α and β. (C) Efficacy of Hsp90 knockdown and expression levels of HSPA1A were verified in total protein extracts. TUBA4A was used as loading control. (D) Cells were treated with sodium arsenite (50 μM) for 45 min, followed by recovery in drug-free medium. Kinetics of SG dissolution are reported. Images were taken over a time period of 4 h every 10 min. Dashed lines = 95% confidence intervals. Number of cells counted: 159 (siRNA control); 166 (siRNA Hsp90 α + β).

E HeLa-Kyoto cells were treated with HS at 43.5°C for 1 h. Cells were then allowed to recover at 37°C for 1 h in drug-free medium (control) or in presence of VER (40 μM), GA (5 μM), or 17AAG (5 μM). Cells were fixed and stained for the SG marker TIA-1, and the percentage of cells with SGs was counted. Number of cells counted: 605 (recovery control); 510 (recovery VER); 1,334 (recovery GA); and 468 (recovery 17AAG). *n* = 3–4 independent experiments, ± s.e.m. *P* < 0.01 (One-way ANOVA).

F G3BP2-GFP HeLa-Kyoto cells were treated with MG132 (20 μM) for 3 h, followed by recovery for 2 h in drug-free medium (control) or in presence of VER (40 μM), GA (5 μM), or 17AAG (5 μM). Cells were fixed, and the percentage of cells with SGs was counted. Number of cells counted: 439 (recovery control); 649 (recovery VER); 637 (recovery GA); and 649 (recovery 17AAG). *n* = 3 independent experiments, ± s.e.m. *P* < 0.0001 (One-way ANOVA).

Data information: Related to Fig EV1 and Movies EV1–EV5.

several protein components and transiently interact, allowing the transfer of specific mRNPs between these two types of granules (Kedersha *et al*, 2005). We thus asked whether inhibition of Hsp90 may perturb the equilibrium between SGs and PBs, indirectly affecting SG dynamics.

To test this hypothesis, we first visualized PBs using an antibody specific for the PB-resident protein DCP1A. In line with previous findings, long-term treatment of HeLa cells with GA or 17AAG for 24 h led to the disappearance of PBs (Fig EV1I), whereas VER had no effect on PB number (Fig EV1I). Treatment of HeLa cells with GA or 17AAG for 4 h affected PB formation only mildly (Fig EV1I). Of note, after a 1 h treatment PBs were not visibly affected, while SG dynamics were very strongly affected (Fig 1A and B).

We next investigated the movements of SGs and PBs. While SGs constantly fuse and divide, PBs maintain their size and shape and move rapidly, often docking onto SGs, a process referred to as kissing (Kedersha *et al*, 2005). Thus, although Hsp90 inhibition may not change the number of PBs, it may affect the dynamic interactions between SGs and PBs, ultimately influencing SG disassembly kinetics. We performed live-cell imaging in cells co-expressing the markers G3BP2-GFP and mRFP-DCP1A to study SG and PB dynamics (Kedersha *et al*, 2008). SGs were induced using sodium arsenite, and SG and PB dynamics were investigated during the recovery phase, in the absence or presence of the Hsp70 or Hsp90 inhibitors. We observed kissing events between PBs and SGs under all conditions tested, regardless of chaperone inhibition (Fig EV1J and Movie EV6). Combined these data support the notion that Hsp90 inhibition does not affect SGs through its effect on PBs, but through a direct mechanism.

## DYRK3 is a new client of Hsp90

Hsp90 fulfills housekeeping functions by assisting the folding and maturation of a large variety of proteins, referred to as Hsp90 clients. Hsp90 clients include proteins known to be recruited into SGs, such as raptor, as well as several protein kinases (Delgoffe *et al*, 2009; Taipale *et al*, 2010; Thedieck *et al*, 2013; Schopf *et al*, 2017; Tsuboyama *et al*, 2018). Association of client proteins with Hsp90 is a prerequisite for their maturation into an activation-competent state; as a result, inhibition of Hsp90 disrupts the

signaling mediated by, e.g., cyclin-dependent kinases (Stepanova *et al*, 1996), Src kinases (Bijlmakers & Marsh, 2000) and the PI3K-AKT cascade (Basso *et al*, 2002) and often leads to the degradation of the clients, causing a decrease in their steady-state levels. Based on this, we hypothesized that Hsp90 may be required to maintain the activity of key factors that regulate SG disassembly.

One candidate is DYRK3, a DYRK family member, which is activated by tyrosine autophosphorylation in the conserved YXY activation loop (Li *et al*, 2002). DYRK3 recently emerged as an important constitutively active kinase that promotes the disassembly of SGs and of other condensates, including SC35 splicing speckles upon entry into mitosis (Wippich *et al*, 2013; Rai *et al*, 2018). Indeed, Hsp90 alpha (HSP90AA1) and beta (HSP90AB1) were identified previously as DYRK3 interactors (Rai *et al*, 2018), suggesting that Hsp90 may act upstream of DYRK3 to regulate SG disassembly (Fig 2A).

We confirmed that Hsp90 interacts with overexpressed GFP-DYRK3 or endogenous DYRK3 and that this interaction is sensitive to inhibition of Hsp90 with GA by immunoprecipitation (Fig 2B) and using proximity ligation assay (PLA) (Figs EV2A and B, and 2C). Using these two techniques, we found that interaction between Hsp90 and DYRK3 is also sensitive to inhibition of DYRK3 kinase activity with the chemical GSK626616 (GSK) (Fig 2D and E) (Wippich *et al*, 2013).

We hypothesized that interaction with Hsp90 may regulate DYRK3 stability in a similar manner as other Hsp90 clients, such as raptor (Delgoffe *et al*, 2009). For raptor, we observed that the steady-state level decreased by 90% in HeLa cells treated with GA or 17AAG, but not with the Hsp70 inhibitor VER (Fig EV2C). For GFP-DYRK3, we found that Hsp90 inhibition reduced the total protein levels by 70% compared to untreated cells. This effect was specific to GFP-DYRK3, since GA and 17AAG only mildly reduced the expression levels of GFP, used as a control (Fig 2F). Also, this effect was specific to Hsp90, since inhibition of Hsp70 with VER did not change GFP-DYRK3 protein levels (Fig 2F). To test whether Hsp90 inhibition promotes GFP-DYRK3 degradation via the proteasome, we co-treated GFP-DYRK3 expressing cells with MG132. GFP-DYRK3 protein levels were partly recovered upon proteasome inhibition, demonstrating that GFP-DYRK3 is unstable and rapidly degraded by the proteasome when Hsp90 is inhibited (Fig 2G).

 

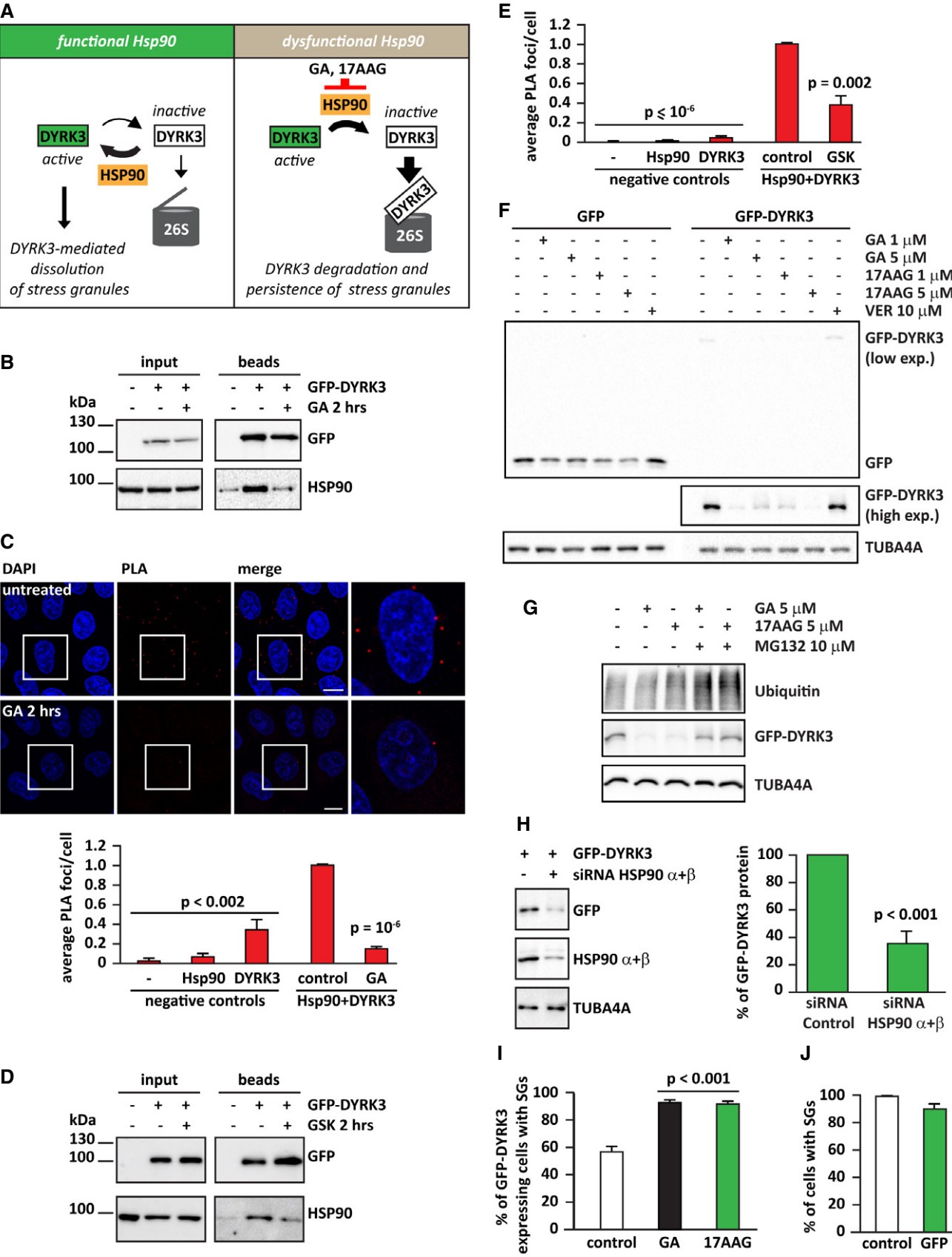

**Figure 2.**

**Figure 2.  DYRK3 is a new client of Hsp90.**

A   Schematic model showing how Hsp90 could regulate DYRK3 stability and activity.

B   HeLa cells were lipofected with a cDNA encoding for an empty vector or GFP-DYRK3. Where indicated, cells were treated for 2 h with GA (5 μM). Cell lysates were subjected to immunoprecipitation of GFP-DYRK3. Levels of GFP-DYRK3 and endogenous Hsp90 are shown in the input and bead fractions.

C   HeLa cells were subjected to proximity ligation assay (PLA) using antibodies specific for endogenous Hsp90 and DYRK3. PLA foci and nuclei were segmented, and PLA foci/cell were automatically quantified. The mean of PLA/foci in cells incubated with no antibodies (−), only Hsp90 antibody, only DYRK3 antibody and both Hsp90 and DYRK3 antibodies is shown. Cells incubated with Hsp90 and DYRK3 antibodies and left untreated were used as control. $n = 3$ independent experiments, $\pm$ s.e.m.; 368–504 cells counted/sample, $P < 0.002$ between the negative controls and cells incubated with Hsp90 and DYRK3 antibodies; $P = 10^{-6}$ between cells incubated with Hsp90 and DYRK3 antibodies and left untreated (control) or treated for 2 h with GA (5 μM) (one-way ANOVA). Scale bar is 10 μm.

D   HeLa cells were lipofected with a cDNA encoding for an empty vector or GFP-DYRK3. Where indicated, cells were treated for 2 h with GSK (5 μM). Cell lysates were subjected to immunoprecipitation of GFP-DYRK3. Levels of GFP-DYRK3 and endogenous Hsp90 are shown in the input and bead fractions.

E   HeLa cells were subjected to proximity ligation assay (PLA) using antibodies specific for endogenous Hsp90 and DYRK3. PLA foci and nuclei were segmented, and PLA foci/cell were automatically quantified. The mean of PLA/foci in cells incubated with no antibodies (−), only Hsp90 antibody, only DYRK3 antibody and both Hsp90 and DYRK3 antibodies is shown. Cells incubated with Hsp90 and DYRK3 antibodies and left untreated were used as control. $n = 3$ independent experiments, $\pm$ s.e.m.; 287–578 cells counted/sample, $P < 10^{-6}$ between the negative controls and cells incubated with Hsp90 and DYRK3 antibodies; $P = 0.002$ between cells incubated with Hsp90 and DYRK3 antibodies and left untreated (control) or treated for 2 h with GSK (5 μM) (one-way ANOVA).

F   HeLa cells were lipofected with a cDNA encoding for GFP or GFP-DYRK3. Six hours post-transfection, cells were incubated in drug-free medium (−) or in presence of GA, 17AAG, or VER (concentrations are shown). Total proteins were extracted 16 h later. GFP and TUBA4A protein levels were analyzed by immunoblotting. Representative immunoblotting of 3 independent experiments.

G   HeLa cells were lipofected with a cDNA encoding for GFP-DYRK3. 16 h post-transfection, cells were incubated for an additional 8 h in drug-free medium (−) or in presence of GA or 17AAG alone or combined with MG132 (concentrations are shown). Total proteins were extracted and GFP, and ubiquitin and TUBA4A protein levels were analyzed by immunoblotting. Representative immunoblotting of 3 independent experiments.

H   HeLa cells were lipofected with non-targeting control siRNA (−) or siRNA pools specific for Hsp90 α and β. 48 h later, cells were lipofected with a cDNA encoding for GFP-DYRK3. 24 h post-transfection, total proteins were extracted and GFP-DYRK3, Hsp90 α, and β, and TUBA4A protein levels were analyzed by immunoblotting. GFP-DYRK3 protein levels were quantified and are shown. $n = 3$ independent experiments, $\pm$ s.e.m.; $P < 0.001$ (Student's $t$-test).

I   HeLa cells were lipofected with a cDNA encoding for GFP-DYRK3. 24 h post-transfection, cells were treated for 45 min with arsenite (50 μM) alone (control) or with GA (5 μM) or 17AAG (5 μM). Cells were then fixed and the percentage of cells expressing GFP-DYRK3 with SGs was counted. Total cells counted: 462 (sodium arsenite); 444 (sodium arsenite and GA); and 561 (sodium arsenite and 17AAG). $n = 3$ independent experiments, $\pm$ s.e.m.; $P < 0.001$ (One-way ANOVA).

J   HeLa cells were either left untreated or lipofected with a cDNA encoding for GFP. 24 h post-transfection, cells were treated for 45 min with arsenite (50 μM). The percentage of cells forming SGs is shown (only GFP-positive cells were counted in the GFP-overexpressing population). Total cells counted: 934 (control) and 555 (GFP). $n = 3$–4 independent experiments, $\pm$ s.e.m. (Student's $t$-test).

Data information: Related to Fig EV2 and Movie EV6.

Similar results were obtained when depleting Hsp90 alpha and beta by siRNA transfection (Fig 2H). Together, these data suggest that DYRK3 requires repeated rounds of binding and release to Hsp90 to maintain its active state; preventing DYRK3 reloading onto Hsp90 redirects it to degradation, similar to what previously found for other client kinases of Hsp90 (Fig 2A) (Taipale *et al*, 2012). This interpretation is further substantiated by live-cell imaging studies with cells expressing GFP-DYRK3. In normally growing cells, GFP-DYRK3 forms condensates depending on the expression level (Fig EV2D), as previously reported (Wippich *et al*, 2013). However, upon inhibition of Hsp90, these GFP-DYRK3 condensates progressively disappear and the cytosolic GFP-DYRK3 signal decreases (Fig EV2D). Given that DYRK3 continuously diffuses into and out of condensates (Wippich *et al*, 2013), these data suggest that prolonged Hsp90 inhibition may induce an unstable conformation in DYRK3, redirecting it to degradation. Thus, reduced DYRK3 levels, due to Hsp90 inhibition, could delay SG disassembly.

Previous work had shown that inhibition of DYRK3 kinase activity with GSK delays SG disassembly (Wippich *et al*, 2013; Rai *et al*, 2018). In agreement, addition of GSK during the recovery phase led to a significant SG persistence (Fig EV2E). Conversely, overexpression of GFP-DYRK3 in cells exposed to arsenite promoted SG disassembly and caused a decrease in the percentage of cells with SGs (Rai *et al*, 2018). We could confirm this result and found that SGs were present in circa 60% of GFP-DYRK3 overexpressing cells that were treated with arsenite (Fig 2I). By contrast, the percentage of SG-positive cells increased to more than 90% upon co-treatment with GA or 17AAG (Fig 2I). The decrease in SG-positive cells was specific to GFP-DYRK3 since cells overexpressing GFP as a control readily formed SGs upon arsenite treatment (Fig 2J).

Finally, to further test the interplay between Hsp90 and DYRK3, by live-cell imaging we compared the impact of GA on SGs disassembly in control versus DYRK3 depleted cells (Fig EV2F and G). Addition of GA during the recovery phase after sodium arsenite treatment delayed SG disassembly less efficiently in DYRK3-depleted cells compared to control cells (Fig EV2G). These data suggest that Hsp90 regulates SG dynamics, at least in part, by targeting DYRK3 stability and activity.

## Inhibition of Hsp90 prevents DYRK3-mediated dissolution of aberrant SC35-mitotic bodies

The results presented above demonstrate that Hsp90 interacts with DYRK3 and affects its stability in cells. To show that Hsp90 also affects DYRK3 function, we monitored the subcellular distribution of DYRK3, as this is influenced by its activity (Rai *et al*, 2018). In resting HeLa cells, DYRK3 is diffusely distributed in the cytoplasm and nucleus, where it is enriched in splicing speckles containing the SC35 splicing factor. Accumulation of DYRK3 inside SC35-speckles increases strongly following inhibition of DYRK3 with GSK (Rai *et al*, 2018; Fig EV3A and B). In mitotic cells, inhibition of DYRK3 with GSK also leads to the formation of aberrant SC35 bodies (Rai *et al*, 2018; Fig EV3C). This suggests that the increased localization of DYRK3 in SC35 splicing speckles and the formation of aberrant mitotic SC35 bodies is a readout for DYRK3 activity in cells.

Accordingly, we determined whether inhibition of Hsp90 affects the subcellular distribution of DYRK3. We found that DYRK3 was diffusely distributed throughout the cytoplasm and nucleus in normally growing cells, with occasional recruitment to SC35 splicing speckles; a similar distribution of DYRK3 was observed in cells treated with VER (Fig 3A and B). By contrast, DYRK3 was enriched in nuclear splicing speckles in cells treated with GA or 17AAG (Fig 3A and B). Similar results were obtained in a HeLa cell line expressing GFP-DYRK3 (Rai et al, 2018), excluding antibody artifacts. Of note, aberrant SC35 bodies assembled in mitotic cells upon Hsp90 inhibition or by depleting Hsp90 alpha and beta through siRNA treatment (Figs 3C–E and EV3D). We conclude that inhibition or downregulation of Hsp90 impairs DYRK3 activity in a similar manner as direct inhibition of DYRK3 with the specific inhibitor GSK.

## DYRK3 partitioning into condensates protects it from aggregation upon Hsp90 inhibition

Although Hsp90 inhibition increased the targeting of DYRK3 to SC35 splicing speckles in interphase cells (Figs 3A and B and EV3A and B), we found no evidence of colocalization between Hsp90 and DYRK3 inside SC35 speckles (Fig EV3E). In addition, we noticed occasional staining of GFP-DYRK3 inside nucleoli (Fig EV3F), which increased upon Hsp90 inhibition (Fig EV3G). Yet, DYRK3 is not required to disassemble nucleoli (Rai et al, 2018). Together these observations raise questions about the functional significance of DYRK3 localization changes upon Hsp90 inhibition.

Upon stress, the nucleolus accumulates misfolding-prone proteins and protects them from irreversible aggregation (Frottin et al, 2019; Mediani et al, 2019). Considering that Hsp90 inhibition leads to DYRK3 destabilization, we wondered whether partitioning into condensates such as SGs, SC35 speckles, and nucleoli may protect DYRK3 from aggregation and/or degradation. To test this

hypothesis, we overexpressed a deletion mutant of GFP-DYRK3 lacking the N-terminal domain (GFP-DYRK3-dN), which is unable to partition into membraneless condensates (Wippich et al, 2013), but still binds Hsp90 (Fig EV3H). GFP-DYRK3-dN was diffusely distributed in the cytoplasm and nucleoplasm of untreated cells (Figs 3F and EV3G, lower panel). However, upon Hsp90 inhibition, GFP-DYRK3-dN did not partition in SC35 speckles (Fig EV3G, lower panel), but it formed aggregate-like structures in the perinuclear region (Fig 3F and Movie EV7). These perinuclear aggregates showed a low protein mobility as determined by Fluorescence Recovery After Photobleaching (FRAP); by contrast, soluble cytosolic and nuclear GFP-DYRK3-dN was highly mobile in untreated cells (Fig 3G). Next, we separated cell lysates expressing WT DYRK3 and its dN variant into NP-40 soluble and insoluble fractions. Fractions were prepared from cells treated with sodium arsenite, which promotes the targeting of WT DYRK3, but not of DYRK3-dN, into SGs (Wippich et al, 2013) and also after recovery in drug-free medium (control) or in presence of GA. The levels of WT DYRK3 were similar in the soluble and insoluble fractions under all conditions tested (Fig 3H). By contrast, DYRK3-dN accumulated in the insoluble fraction upon inhibition of Hsp90 (Fig 3H).

Finally, we replaced the N-terminus of DYRK3 with the NM domain of the yeast Sup35 protein, which is known to target proteins into SGs (Gilks et al, 2004). We then studied the targeting of this chimeric protein, referred to as Sup35NM-DYRK3-dN, to SGs and its aggregation propensity upon Hsp90 inhibition. Sup35NM-DYRK3-dN was strongly recruited inside SGs (Fig EV3I). Importantly, upon treatment of the cells with GA, Sup35NM-DYRK3-dN was diffusely distributed in the cytoplasm (Fig EV3J) and its expression levels progressively decreased (Fig EV3J and K), similar to what observed for DYRK3-WT (Fig 2F). In addition, when GA was added during the recovery phase after arsenite treatment, we found that Sup35NM-DYRK3-dN was sequestered inside persisting SGs, while DYRK3-dN formed perinuclear aggregates (Fig 3I). One

---

**Figure 3.** **Inhibition or depletion of Hsp90 relocalizes DYRK3 to splicing speckles and leads to the accumulation of aberrant SC35 mitotic bodies.**

A, B   HeLa cells were left untreated (control) or treated with GA (5 μM), 17AAG (5 μM) or VER (10 μM) for 8 h. Cells were then fixed and immunostained for the nuclear speckle marker SC35 and DYRK3. Automatic segmentation of nuclear speckles is based on SC35. An automated imaging assay was used to quantify the percentage of nuclear speckles highly enriched for DYRK3 (using a fluorescent ratio 1.5 as threshold). Scale bar is 10 μm. (B) Number of speckles quantified = 1,807 (control); 5,110 (GA); 3,596 (17AAG); 1,556 (VER); $P < 10^{-10}$, ± s.e.m. (one-way ANOVA).

C, D   Representative confocal microscopy images of cells with SC35-positive bodies (D) and quantification of the percentage of mitotic cells showing aberrant SC35 mitotic bodies is reported (C). Number of cells counted: 923 (control), 694 (GA), 951 (17AAG), and 1,019 (VER). $n = 6$ independent experiments, ± s.e.m., $P < 10^{-10}$, ± s.e.m. (one-way ANOVA). (D) Scale bar is 10 μm.

E   HeLa cells were lipofected with non-targeting control siRNA or siRNA pools specific for Hsp90 α and β. 72 h post-transfection, cells were fixed and stained for DYRK3 and SC35. Quantification of the percentage of mitotic cells showing aberrant SC35 mitotic bodies is reported. Number of cells counted: 644 (control); 654 (siRNA Hsp90 α and β). $n = 3$ independent experiments, ± s.e.m.; $P = 0.01$ (Student's t-test).

F   HeLa cells were lipofected with a cDNA encoding for GFP-DYRK3-dN. 24 h post-transfection GFP-DYRK3-dN cells were either left untreated or exposed to GA (5 μM) for 4 h. In untreated cells (control), GFP-DYRK3-dN is diffusely distributed in the cytosol and in the nucleus. Upon GA treatment, GFP-DYRK3-dN forms perinuclear (PN) aggregates. Representative confocal images are shown. Scale bar is 10 μm.

G   HeLa cells expressing GFP-DYRK3-dN were treated as described in F. GFP-DYRK3-dN PN aggregates were already visible 2 h after treatment with GA. GFP-DYRK3-dN mobility was investigated by FRAP in the cytoplasm and nucleus of untreated cells, and in the PN aggregates of GA-treated cells. The mean of 9–11 FRAP curves for each condition and the fitting curves are shown; s.e.m. is shown.

H   HeLa cells were lipofected with cDNAs encoding for GFP-DYRK3 or GFP-DYRK3-dN. 24 h post-transfection cells were treated with sodium arsenite (A) to induce SGs and 45 min after they were allowed to recover for 4 h in drug-free medium (rec. control) or in presence of GA (5 μM; rec. GA). NP-40 soluble and insoluble proteins were fractionated and the distribution of GFP-DYRK3 and GFP-DYRK3-dN in both fractions was analyzed by immunoblotting. TUBA4A was used as loading control. Quantitation of the ration between GFP NP40 soluble and insoluble levels is shown; $n = 3$ independent experiments, ± s.e.m. (Student's t-test).

I   mCherry-G3BP1 expressing HeLa cells were lipofected with cDNAs encoding for GFP-DYRK3-dN or Sup35-NM-GFP-DYRK3-dN. 24 h post-transfection, cells were treated for 45 min with sodium arsenite to induce SGs, followed by recovery for 4 h in presence of GA (5 μM; rec. GA). Representative confocal microscopy images are shown. Nucleic acid was stained with DAPI. Scale bar is 10 μm.

Data information: Related to Fig EV3 and Movie EV7.

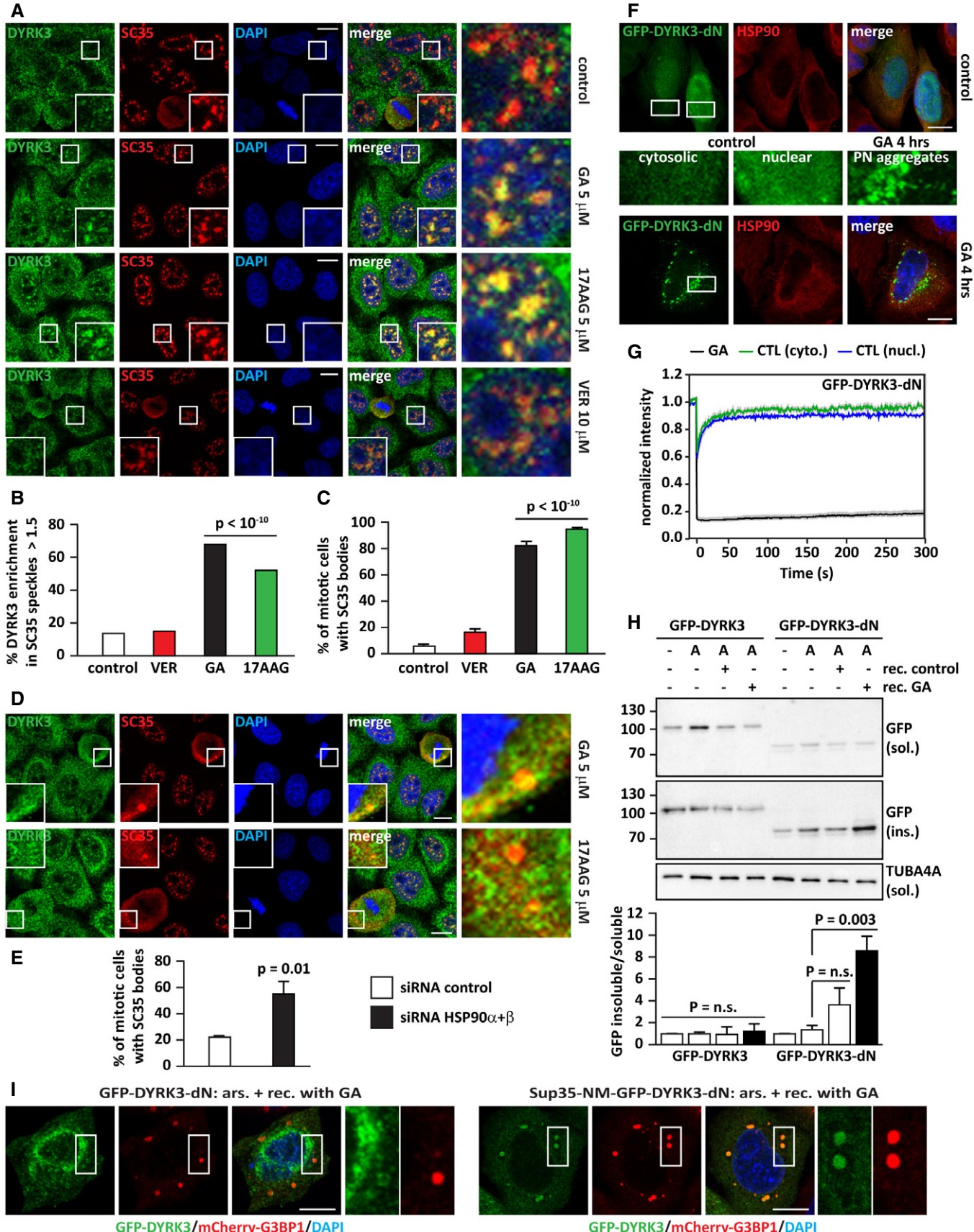

**Figure 3.**

possible explanation is that DYRK3 adopts an aggregation-prone metastable state in the absence of Hsp90 and that condensate targeting protects DYRK3 from irreversible aggregation.

## Hsp90 regulates DYRK3 stability and activity outside/at the boundary of condensates

Upon DYRK3 inhibition SGs persist for longer time (Wippich et al, 2013; Rai et al, 2018). We confirmed that DYRK3 is present in normal SGs and persisting SGs that failed to disassemble due to inhibition of Hsp90 or inhibition of DYRK3 (Figs 4A and EV4A). Of note, there was no colocalization of Hsp90 with DYRK3 inside SGs (Figs 4A and EV4A), similar to what we observed for SC35 speckles. We found no evidence of colocalization of Hsp90 with cytoplasmic DYRK3 condensates (Fig 4B), but we occasionally detected a ring of Hsp90 around them (Fig 4B, lower panel). Moreover, proximity ligation assay (PLA) foci indicative of a Hsp90/DYRK3 interaction were generally localized outside of DYRK3 condensates (Fig EV4B). In addition, DYRK3 interaction with Hsp90 decreased following arsenite treatment, which promotes the partitioning of DYRK3, but not of Hsp90, inside SGs. These data suggest that Hsp90 interacts with non-condensed DYRK3 and regulates its stability and activity outside or at the boundary of condensates (see also Fig EV2D).

This raises the question whether Hsp90-free DYRK3 inside SGs can promote SG disassembly or whether DYRK3 has to associate with Hsp90 outside or at the boundary of SGs to engage with SG proteins and promote the SG disassembly process.

We tested this using fluorescence recovery after photobleaching (FRAP). Indeed, GFP-DYRK3 was able to associate with SGs in the recovery phase (Fig 4C, upper panel). Importantly, inhibition of Hsp90 with GA (Fig 4C, middle panel) or inhibition of DYRK3 with GSK did not affect DYRK3 recruitment to SGs (Fig 4C, lower panel), although SG disassembly was delayed under these conditions (Figs 1A and EV2E). Together these data suggest that DYRK3 associates with Hsp90 outside or at the boundary of SGs and that its active form is required to initiate the disassembly process.

## Hsp90 couples SG disassembly and mTORC1 signaling via DYRK3

DYRK3 couples SG dynamics to the mTORC1 signaling pathway (Wippich et al, 2013), which controls metabolic processes such as mRNA biogenesis and protein synthesis (Abraham, 2002; Ma & Blenis, 2009). Upon stress, the mTORC1 component raptor is sequestered inside SGs (Thedieck et al, 2013). Raptor is released

from SGs in the stress recovery phase, when DYRK3 promotes SG disassembly; moreover, DYRK3 phosphorylates and inactivates PRAS40, an inhibitor of mTORC1, thus enabling translation restoration and cell growth (Wippich et al, 2013). As a consequence, SG disassembly coincides with restoration of translation (Mazroui et al, 2007; Lian & Gallouzi, 2009).

Like DYRK3, Akt/PKB (protein kinase B) phosphorylates PRAS40 at Thr246, regulating the mTORC1 activity (Nascimento et al, 2010; Saxton & Sabatini, 2017). Moreover, Akt is a client of Hsp90, whose stability is reduced upon Hsp90 inhibition (Basso et al, 2002; Ohji et al, 2006; Giulino-Roth et al, 2017). We thus asked whether during the recovery phase after stress, Hsp90 promotes SG disassembly and translation restoration via DYRK3 or via the PI3K/Akt pathway, or both.

To test this idea, we first determined whether inhibition of the PI3K/Akt pathway affects SG dynamics. Treatment of the cells with the inhibitors LY294002 or wortmannin did not promote SG assembly (Movie EV8), nor did it affect the disassembly of SGs (Movie EV9), although the levels of phosphorylated PRAS40 were efficiently decreased (Huang & Porter, 2005) (Fig EV5A and B). Thus, under the conditions tested, inhibition of the PI3K/Akt pathway has no impact on SGs.

Next, we wanted to investigate how SG disassembly is coupled to restoration of translation. Raptor is recruited together with inactive DYRK3 inside SGs (Figs EV5C and EV4A). Conditions that delay SG disassembly such as inhibition of DYRK3, impair the release of raptor, delaying the restoration of the mTORC1 signaling pathway and of translation (Wippich et al, 2013). By acting upstream of DYRK3, Hsp90 inhibition mimics the effects of DYRK3 inhibition (Figs EV5D and EV4A). To investigate in detail how translation restoration depends on SG disassembly, we induced SGs and prepared protein extracts either immediately after treatment or after recovery in the presence or absence of GA or 17AAG. We used 17AAG at two concentrations: 0.5 μM, which delays but does not completely prevent SG dissolution, and 5 μM, which leads to SG persistence in nearly 100% of the cells (see Fig 1B and Movie EV2). We then determined the phosphorylation status of PRAS40, a target of DYRK3, and p70 S6K, a target of mTORC1 kinase. The phosphorylation status of both proteins correlated with the extent of SG disassembly and phosphorylation was barely detectable after 4 h of recovery in the presence of 5 μM 17AAG (Fig EV5E). Importantly, treatment of the cells with GA or 17AAG alone had no effect on the phosphorylation status of PRAS40 or p70 S6K (Fig EV5F). Similar observations were made for another mTORC1 substrate that is also

**Figure 4. Hsp90 regulates DYRK3 activity by acting outside of condensates.**

A  G3BP1-mCherry HeLa-Kyoto cells were left untreated or treated with sodium arsenite (50 μM) for 45 min to induce SGs. Cells were then fixed and immunostained for DYRK3 and total Hsp90 antibodies. Quantitation of the percentage of SGs that show DYRK3 and Hsp90 enrichment (> 1.5). Automated imaging and SG segmentation are based on mCherry-G3BP1 signal. Number of SGs analyzed: 3,171 (arsenite). Scale bar is 10 μm.

B  Confocal microscopy of HeLa-Kyoto cells lipofected for 24 h with cDNAs encoding for GFP-DYRK3 and either an empty vector or mCherry-Hsp90 α and β. The upper panel shows endogenous Hsp90, stained with a specific antibody for total Hsp90; the lower panel shows exogenous mCherry-Hsp90 α and β. Scale bar is 10 μm.

C  Quantitation of the fluorescence intensity recovery after bleach of GFP-DYRK3 in G3BP1-mCherry HeLa-Kyoto cells 24 h after transfection. Cells were treated with sodium arsenite (50 μM) for 45 min to induce SGs, which were visualized with G3BP1-mCherry; then, sodium arsenite was removed and the cells were incubated in drug-free medium or in presence of GA (5 μM) or of GSK (5 μM). FRAP was performed during the stress recovery period. A representative image of cells treated with arsenite followed by recovery in drug-free medium is shown. Arrowheads and dotted circle indicate the ROI. Scale bar is 5 μm. The mean of 13 FRAP curves (recovery control), 12 FRAP curves (recovery GA), and 14 FRAP curves (recovery GSK) is shown in red; the s.e.m. is shown in gray.

Data information: Related to Fig EV4.

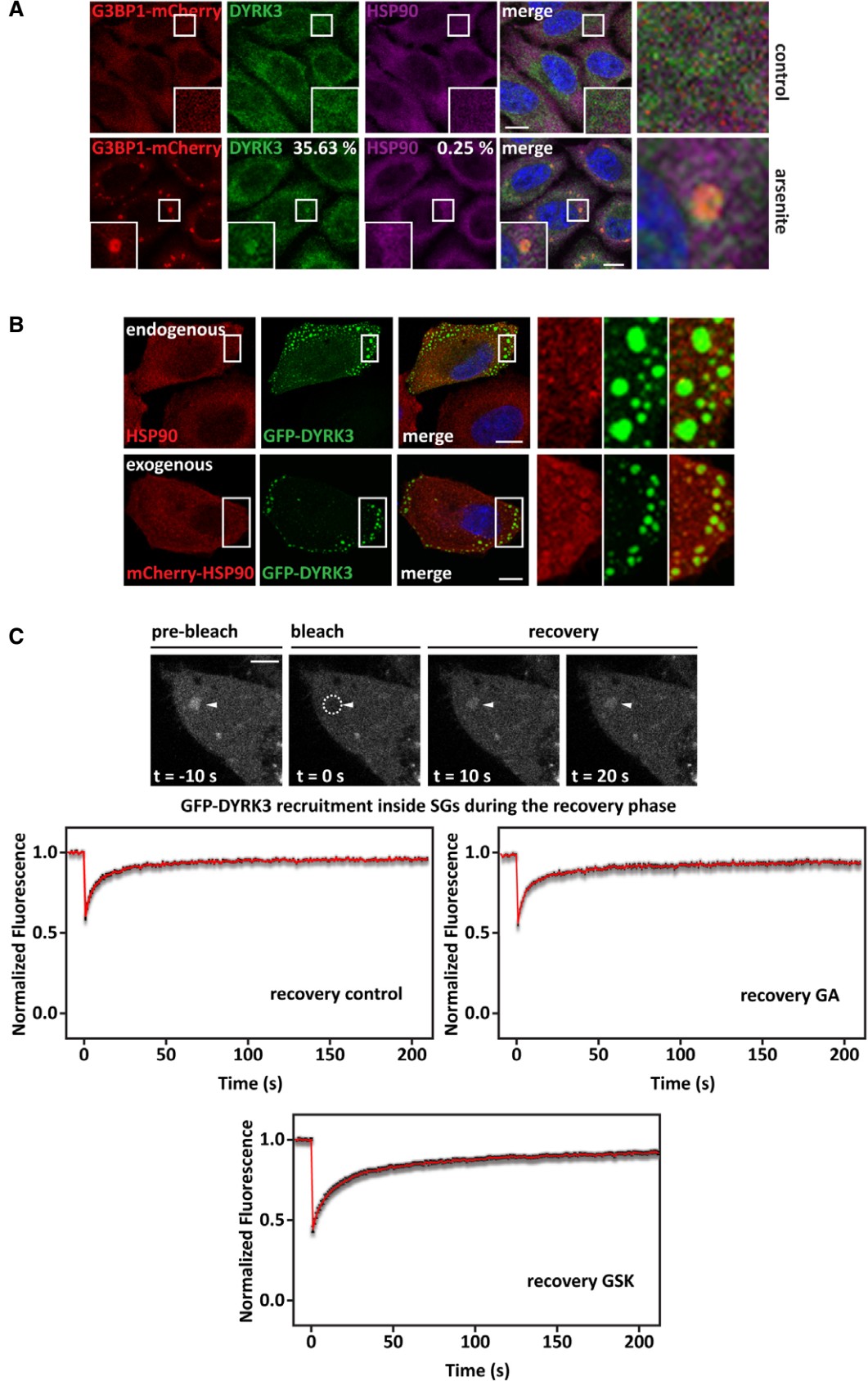

**Figure 4.**

involved in translation regulation: 4E-binding protein 1 (4E-BP1) (Fig EV5E and F) (Saxton & Sabatini, 2017).

To further test the functional connection between Hsp90, SG disassembly, and translation restoration, we treated the cells as described above, but during the last 15 min we incubated the cells with puromycin. Puromycin is incorporated into newly synthesized proteins, thereby allowing us to monitor translation. Inhibition of unstressed cells with GA or 17AAG did not impair puromycin incorporation (Fig EV5G), in agreement with the observation that there was no induction of SGs (Fig EV1B), nor changes on the phosphorylation status of PRAS40 (Fig EV5F). By contrast, arsenite treatment decreased puromycin incorporation (Fig EV5G). Importantly, in the stress recovery phase, the amount of puromycylated proteins correlated with the extent of disassembled SGs (Fig 1A and B, and Movie EV2) and the levels of phosphorylated PRAS40, p70 S6K, and 4E-BP1 (Fig EV5E). Puromycin incorporation was detected already after 1 h of recovery in drug-free medium (Fig EV5G), which allowed SG disassembly in nearly all the cells (Fig 1A and Movie EV2), but not in presence of GA or 17AAG, which led to SG persistence in nearly 80% of the cells (Fig 1A and B and Movie EV2). After 4 h of recovery, protein synthesis was not restored in cells treated with a high 17AAG concentration that caused SG persistence in more than 90% of the cells (Figs EV5G and 1B and Movie EV2).

Together, these data suggest that Hsp90 promotes translation restoration through the reactivation of two translation-regulatory kinases, DYRK3, and mTORC1, which are gradually released from disassembling SGs. In addition, Hsp90 affects translation restoration through the PI3K/Akt pathway (Ohji et al, 2006; Giulino-Roth et al, 2017), but this pathway is independent of SGs. Thus, cells coordinate translation restoration after stress by activating two synergistic pathways, one SG-dependent and one independent, both of which are highly sensitive to Hsp90 activity (Fig EV5H).

### HeLa cells and iPSC-MNs expressing mutated FUS, as well as ALS patients' fibroblasts and spinal cord MNs are vulnerable to Hsp90 inhibition and show defective induction of DYRK3

Maturation of SGs into pathological inclusions is emerging as an important pathomechanism in ALS, which is characterized by the progressive loss of motor neurons (MNs) (Nedelsky & Taylor, 2019). Several important factors have been identified to date that affect SG conversion into an aberrant state: Hsp70, VCP, and the autophagy receptor p62/SQSTM1 (Alberti & Carra, 2018; Zhang et al, 2019). However, the role of Hsp90 in ALS pathogenesis is still unclear. Interestingly, a recent study demonstrated that inhibition of Hsp90 with GA triggers ALS MN death and that the sensitivity of primary MNs to GA is 100-time higher than that of other types of neurons (Strayer et al, 2019). Although the increased vulnerability of motor neurons to Hsp90 inhibition has been partly ascribed to the inhibition of the Akt pro-survival pathway, our data suggest the possibility that it could be linked to SG persistence, reduced DYRK3 and mTORC1 activity and insufficient levels of translation.

To test this idea, we studied whether Hsp90 inhibition affects SG dynamics in MNs. We used WT and P525L FUS-eGFP induced pluripotent stem cells (iPSC), which were differentiated into MNs to model ALS (Marrone et al, 2018). Compared with MNs expressing WT FUS-eGFP, MNs expressing P525L FUS-eGFP were more sensitive to sodium arsenite treatment and formed SGs,

which were identified by co-labeling with an antibody specific for the TIAR protein, used as a SG marker (Fig 5A and B). Next, we studied the disassembly of SGs in P525L FUS-eGFP MNs in the absence and presence of the Hsp90 inhibitor 17AAG, which has a lower toxicity than GA (Schulte & Neckers, 1998), as well as in presence of the DYRK3 inhibitor GSK. Hsp90 and GSK inhibition both significantly delayed SG dissolution in P525L FUS-eGFP MNs compared to cells that were allowed to recover in drug-free medium (Fig 5C).

Comparable results were obtained with HeLa cell lines that stably express GFP-tagged WT FUS or a corresponding ALS-associated variant of FUS (G156E) (Ganassi et al, 2016). GA had a stronger impact on SG disassembly in the FUS G156E cell line, which also displayed an overall increase of the percentage of persistent SGs compared to control cells (Fig 5D and Movie EV10). In addition, silencing Hsp90 delayed SG disassembly in HeLa cells expressing GFP-FUS-WT and G156E, with a slightly stronger impact in the latter ones (Fig 5E). Next, we tested the sensitivity of these cell lines to chemical inhibition of DYRK3. We found that incubation of the cells with GSK during the recovery phase after arsenite treatment delayed SG disassembly in both GFP-FUS-WT and G156E expressing HeLa cells (Fig 5F). Together, these data indicate that impaired SG disassembly because of Hsp90 or DYRK3 inhibition affects cells expressing ALS-linked protein variants.

Next, we used fibroblasts from three healthy donors, from one sporadic ALS (sALS) patient, and from one familiar ALS (fALS) patient, carrying the p.R191Q mutation in the gene coding for the chaperone VCP (Johnson et al, 2010). We studied the expression levels of constitutive HSP90AA1, stress-inducible HSP90AB1 and DYRK3 in these fibroblast lines after treatment with arsenite, to induce SGs, and after the recovery phase. HSP90AA1, HSP90AB1, and DYRK3 were all significantly upregulated in control fibroblasts in the recovery phase (Fig 5G). sALS and fALS fibroblasts upregulated HSP90AA1 and HSP90AB1 similar to control fibroblasts (Fig 5H); by contrast, they could not induce DYRK3 as efficiently as control fibroblasts after stress dissipation, when DYRK3 activity is required to disassemble SGs (Fig 5H). Thus, deregulated expression and function of Hsp90 and DYRK3 may contribute to the impaired cell stress response and altered SG dynamics observed in ALS.

Finally, to better understand the role of DYRK3 in ALS pathogenesis, we used autopsy tissue from FUS-fALS patients, as well as from normal controls. At first, by using DYRK3 and FUS antibodies we assessed the localization of DYRK3 and FUS in lumbar spinal cord alpha-motor neurons (α-MNs) from FUS-ALS patients and compared them with the normal controls. In control α-MNs, as expected, a diffuse pattern of nuclear FUS immunoreactivity was observed, while DYRK3 protein was uniformly distributed throughout the cytoplasm (Fig 5I, upper panel, white arrows); DYRK3 also showed strong speckled pattern of immunoreactivity in the nucleus (Fig 5I, upper panel, white arrowhead). Such speckled pattern of nuclear immunoreactivity of DYRK3 rarely colocalized with nuclear FUS immunoreactivity. Interestingly, surviving α-MNs harboring larger FUS aggregates in FUS-ALS lumbar spinal cord (Fig 5I, yellow arrows) displayed markedly reduced DYRK3 immunoreactivity both in the cytoplasm (Fig 5I, red arrowheads and Fig 5J), as well as in the nucleus (Fig 5I, red arrowheads and Fig 5J). In contrast, adjacent α-MNs lacking FUS aggregates in FUS-ALS showed DYRK3

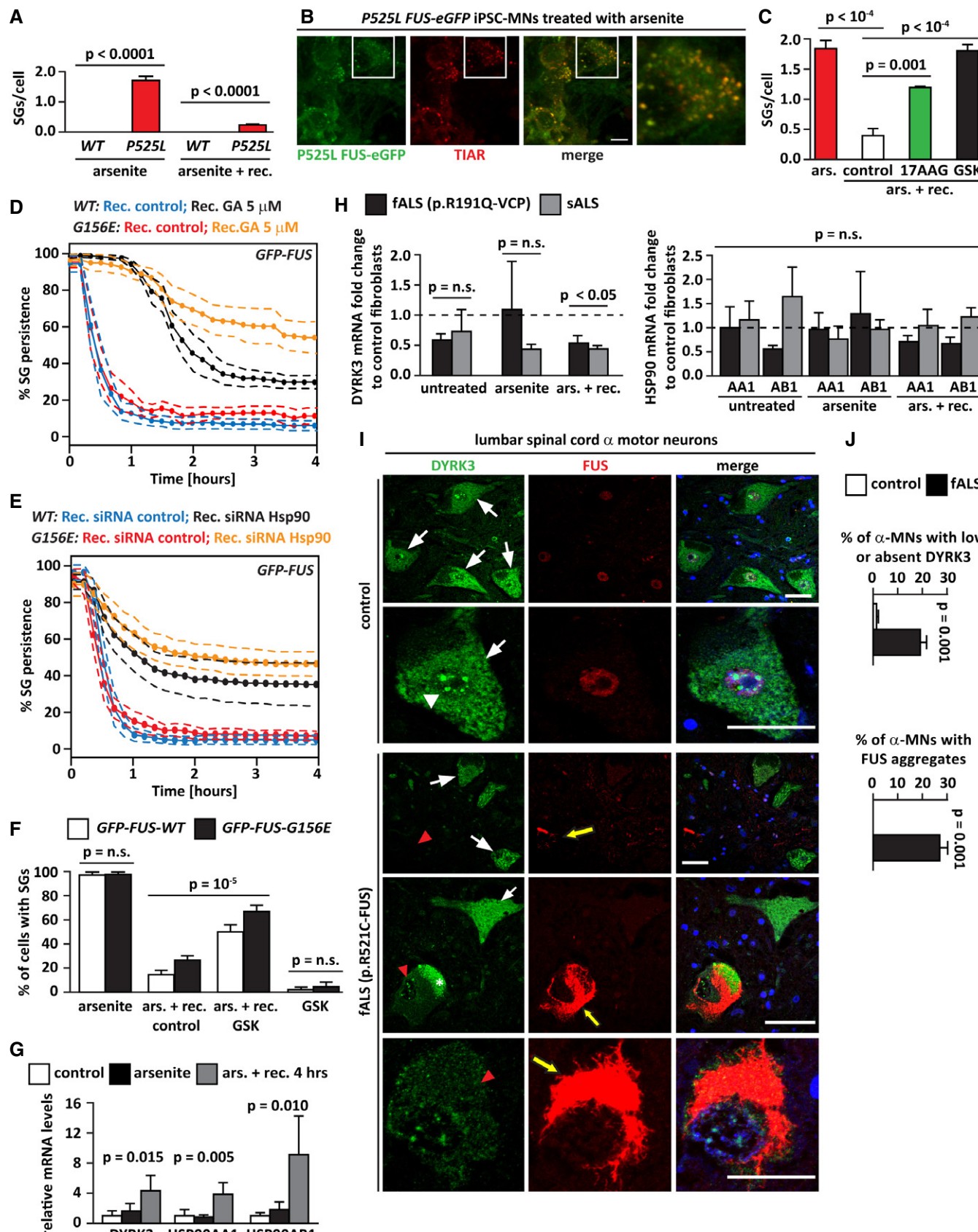

Figure 5.

**Figure 5.  ALS cells are sensitive to Hsp90 inhibition and show decreased DYRK3 expression.**

A   WT and P525L FUS-eGFP iPSCs were differentiated into MNs and were treated with sodium arsenite (500 µM) for 2 h. MNs were fixed and the number of SGs per cell was counted. WT FUS-eGFP iPSC-MNs have no SGs. Number of P525L cells counted: 888 (arsenite); 1,119 (arsenite + rec.). Number of SGs in P525L cells: 1,498 (arsenite); 222 (arsenite + rec.). $n = 6$, $\pm$ s.e.m. (Student's $t$-test).

B   Confocal microscopy images of P525L FUS-eGFP MNs treated with sodium arsenite for 2 h and showing colocalization of FUS with TIAR-positive stress granules. Scale bar is 10 µm.

C   P525L FUS-eGFP MNs were treated with sodium arsenite for 2 h, followed by recovery in drug-free medium (+rec. control) or in presence of 17AAG (30 µM; +rec. 17AAG) or of GSK (5 µM; +rec. GSK) for 6 h. Quantitation of the number of SGs per cell in P525L FUS-eGFP MNs. Number of P525L cells counted and $n$/condition: 2,488, $n = 13$ (arsenite); 1,045, $n = 12$ (arsenite + rec. control); 646, $n = 6$ (arsenite + rec. 17AAG); 924, $n = 3$ (arsenite + rec. GSK), $\pm$ s.e.m. (One-way ANOVA).

D   HeLa cells expressing GFP-tagged wild-type (WT) FUS or the ALS-associated mutant G156E were treated with sodium arsenite (50 µM) for 45 min to induce SGs, followed by recovery in drug-free medium (Recovery Control) or in presence of GA (5 µM). Images were taken over a time period of 4 h every 10 min. Dashed lines = 95% confidence intervals. Number of cells counted: GFP-FUS WT cells, 227 (Recovery Control) and 254 (GA 5 µM); GFP-FUS G156E cells, 149 (Recovery Control) and 177 (GA 5 µM).

E   HeLa cells expressing GFP-tagged wild-type (WT) FUS or the ALS-associated mutant G156E were lipofected with non-targeting control siRNA or siRNA pools specific for Hsp90 α and β. 72 h post-transfection, cells were treated with sodium arsenite (50 µM) for 45 min to induce SGs, followed by recovery in drug-free medium. Images were taken over a time period of 4 h every 10 min. Dashed lines = 95% confidence intervals. Number of cells counted: GFP-FUS WT cells, 532 (siRNA Control) and 449 (siRNA Hsp90); GFP-FUS G156E cells, 566 (siRNA Control) and 504 (siRNA Hsp90).

F   HeLa cells expressing GFP-tagged wild-type (WT) FUS or the ALS-associated mutant G156E were treated with sodium arsenite (500 µM) for 45 min to induce SGs, followed by recovery in drug-free medium (ars. + rec. control) or in presence of GSK (5 µM). Treatment of the cells with GSK (5 µM) for 4 h was included as a control. Cells were fixed, and the % of SG-positive cells was counted. Number of GFP-FUS WT cells counted: 641 (control); 726 (ars.); 846 (ars. + rec. control 4 h); 1,010 (ars. + rec. GSK 4 h); and 422 (GSK 4 h). Number of GFP-FUS G156E cells counted: 538 (control); 602 (ars.); 737 (ars. + rec. control 4 h); 922 (ars. + rec. GSK 4 h); and 424 (GSK 4 h). $n = 4–9$ independent samples, $\pm$ s.e.m.; $P = 10^{-5}$ (between recovery control and recovery with GSK) (one-way ANOVA).

G   DYRK3 and HSP90AA1, HSP90AB1 mRNA levels in three fibroblast lines from healthy individuals. Fibroblast lines were either left untreated (control) or exposed to sodium arsenite (500 µM) for 45 min (arsenite); where indicated, cells were allowed to recover in drug-free medium for 4 h prior to RNA extraction (arsenite + recovery 4 h). $n = 3$ independent experiments, $\pm$ s.e.m. (one-way ANOVA).

H   Fibroblasts from one sporadic ALS (sALS) patient and from one familiar ALS (fALS) patient, carrying the p.R191Q mutation in the VCP gene were treated as described in G. Fold change of DYRK3, HSP90AA1 (AA1), and HSP90AB1 (AB1) mRNA levels compared to the average of the three fibroblast lines from healthy individuals (control) are shown. $n = 3$ independent experiments, $\pm$ s.e.m. (one-way ANOVA).

I   Double immunofluorescence labeling using antibodies against DYRK3 and FUS in lumbar spinal cord α-MNs of healthy subjects (control) and fALS patients carrying the p.521C mutation in the FUS gene. Control (upper panel): α-MNs showing uniform cytoplasmic (white arrows) and speckled pattern of strong nuclear immunoreactivity (white arrowhead) of DYRK3, as well as diffuse nuclear FUS immunoreactivity. fALS (p.R521C-FUS; lower panel): surviving α-MNs harboring large FUS aggregates (yellow arrows) showed markedly reduced DYRK3 immunoreactivity both in the cytoplasm as well as in the nucleus (red arrowheads). Instead, α-MNs devoid of FUS aggregates (white arrows) showed normal DYRK3 immunoreactivity similar to the one of α-MNs from normal controls. Asterix (*) represents non-specific lipofuscin granules in one of the α-MN in FUS-ALS. Paraffin sections; scale bars is 50 µm.

J   Quantification of α-MNs showing low or absent DYRK3 staining (upper graphic) and FUS aggregates (lower graphic) in lumbar spinal cord from five healthy subjects (control) and five fALS patients carrying the p.521C mutation in the FUS gene. Total number of α-MNs analyzed: 403 (control); 132 (fALS). $n = 5$, $\pm$ s.e.m. (Student's $t$-test).

Data information: Related to Movie EV10.

---

immunoreactivity similar to the one of α-MNs in normal controls (Fig 5I, white arrow, middle panel).

Altogether these data support the notion that reduced expression of DYRK3 may contribute to impair SG disassembly in cells derived from ALS patients.

# Discussion

Hsp90 is an evolutionarily conserved molecular chaperone that ensures the folding, stability, and activity of a wide range of clients, including kinases, steroid hormone receptors, signaling molecules, and proteins involved in RNA metabolism (Taipale *et al*, 2010; Schopf *et al*, 2017). A large body of evidence has established the essential role of Hsp90 for protein folding both in non-stressful and stressful conditions (Schopf *et al*, 2017). Here, we describe a new function of Hsp90 in regulating the disassembly of SGs in interphase and of SC35 speckles during mitosis. This new function of Hsp90 is, at least in part, mediated by DYRK3, a kinase that promotes the disassembly of these two types of condensates (Wippich *et al*, 2013; Rai *et al*, 2018). We show that Hsp90 interacts with DYRK3 and regulates its stability. By destabilizing DYRK3 and preventing its reactivation, Hsp90

inhibition leads to the accumulation of aberrant mitotic SC35 bodies and delays the disassembly of SGs during the stress recovery phase. Both events can have profound consequences for cellular function, compromising mitosis and translation restoration when the stress subsides. Hsp90 clients include a number of cell cycle regulators and Hsp90 inhibition is known to cause cell cycle arrest (Whitesell & Lindquist, 2005). Thus, during mitosis, inhibition of DYRK3-mediated dissolution of SC35 speckles may contribute to the cell cycle arrest observed upon Hsp90 inhibition. During stress, transcription and translation of housekeeping genes are attenuated presumably to reduce the load of unfolded proteins on the protein quality control machinery (Morimoto, 1998; Pakos-Zebrucka *et al*, 2016; Hershey *et al*, 2019; Muhlhofer *et al*, 2019). This is in part achieved by reorganizing intracellular components into discrete compartments by the process of phase separation (Gomes & Shorter, 2019). For instance, the formation of SGs is thought to serve a dual function: it protects proteins and RNAs from irreversible aggregation and degradation during stress and it ensures that RNAs are released again in the stress recovery phase, when cells need to reactivate transcription and translation to restore cell functionality (Kedersha & Anderson, 2002; Gilks *et al*, 2004; Wallace *et al*, 2015; Riback *et al*, 2017; Franzmann *et al*, 2018; Kroschwald *et al*, 2018; Guillen-Boixet *et al*, 2020).

SGs act as signaling hubs because, in addition to translation factors and mRNAs, they recruit enzymes and signaling molecules such as raptor and mTOR, which are two key components of the mTORC1 complex (Kedersha *et al*, 2013; Saxton & Sabatini, 2017). Thus, Hsp90 emerges as a critical stress-dependent regulator of cellular processes controlling growth, metabolism, and cell division, by chaperoning soluble cytosolic proteins and also by chaperoning SG dynamics in interphase and SC35 speckle disassembly during mitosis. Since Hsp90 interacts with 60% of the human kinome (Taipale *et al*, 2010), we cannot exclude that other Hsp90 client kinases participate in the process of SG disassembly and translation restoration. Yet, DYRK3-depletion diminished the impact of GA on SG disassembly (Fig EV2G), clearly identifying DYRK3 as one of the Hsp90 targets involved in the regulation of SG dynamics. Furthermore, Hsp90 was recently shown to stabilize several SG components (O'Meara *et al*, 2019). This finding suggests that defective SG disassembly upon Hsp90 inhibition could also result from the destabilization of other yet unknown SG components.

Like other DYRK family members, DYRK3 is activated by tyrosine autophosphorylation in the conserved YXY activation loop (Li *et al*, 2002). Phosphorylation of the intrinsically disordered

N-terminal domain of DYRK3 has been proposed to keep DYRK3 in the cytosol, whereas loss of phosphorylation would promote its partitioning into condensates (Wippich *et al*, 2013). Thus, DYRK3 was proposed to enter into SGs in its inactive and unphosphorylated form via its N-terminal domain (Wippich *et al*, 2013). Yet how DYRK3 re-acquires its kinase activity, which is necessary for SG disassembly, has remained unsolved. We show that DYRK3 associates with Hsp90 outside or at the boundary of SGs to be stabilized. Of note, the boundaries of ribonucleoprotein assemblies such as SGs and PBs are not well-defined, with translation factors being present outside of SGs in close proximity to the condensate interface (Tauber *et al*, 2020); in addition, recent evidence suggests that translation of mRNAs can take place at the boundary of SGs (Mateju *et al*, 2020) and PBs (Davidson *et al*, 2016). Based on our data, we propose the following model: during stress, DYRK3 dissociates from Hsp90 and partitions inside SGs, where it is protected from irreversible aggregation and degradation. During the recovery phase after stress, soluble DYRK3 associates with Hsp90 outside or at the boundary of SGs to initiate the disassembly process (Fig 6, upper part). Chaperoning by Hsp90 in the surrounding cytosol would stabilize otherwise metastable states of DYRK3, similar to what has been previously reported for other intrinsically unstable kinases that

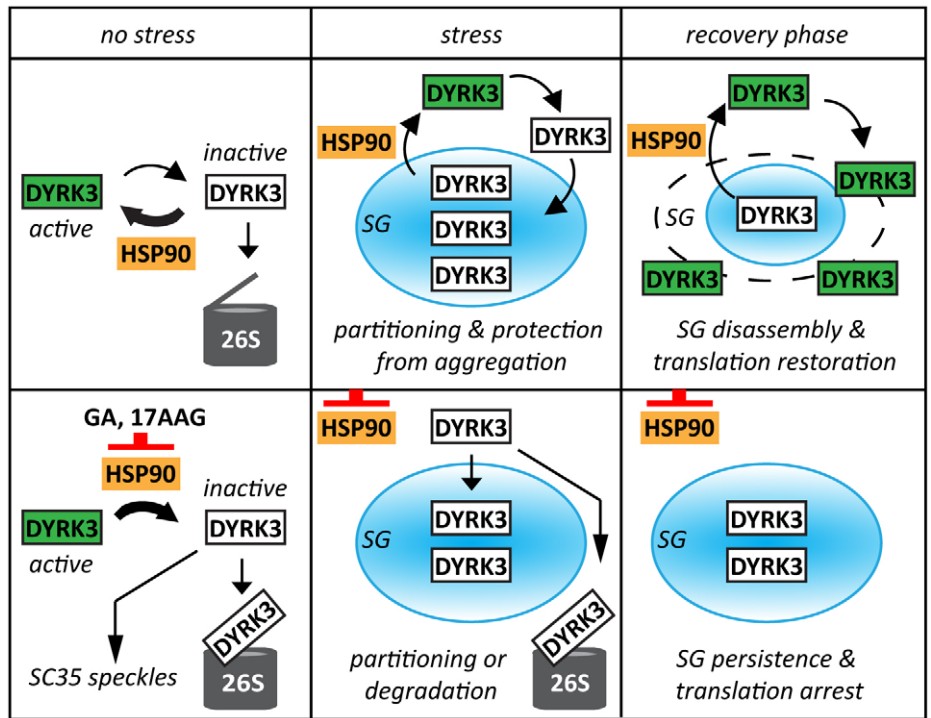

**Figure 6.  Schematic model showing how Hsp90 regulates SG dynamics via DYRK3.**

Upper part: in resting cells (no stress), Hsp90 interacts with DYRK3 and regulates its stability and activity. Upon stress, DYRK3 dissociates from Hsp90 and partitions inside SGs, where it is protected from irreversible aggregation and degradation. During the recovery phase after stress, soluble DYRK3 associates with Hsp90 outside or at the boundary of SGs to be stabilized. Once stabilized by Hsp90, DYRK3 could autophosphorylate its activation loop to achieve full activation and trigger the disassembly of SGs, by phosphorylating yet unknown molecular targets. Active DYRK3 would then promote translation restoration. Lower part: upon treatment with GA or 17AAG, DYRK3 dissociates from Hsp90, becomes inactive and is either targeted to SC35 speckles or degraded by the 26S proteasomes. During stress, blocking the loading of DYRK3 onto Hsp90 with GA or 17AAG redirects DYRK3 to SGs or to 26S proteasomes. During the recovery phase after stress, inhibition of Hsp90 prevents the stabilization and activation of DYRK3. As a consequence, SGs do not disassemble and translation is not restored. The model only depicts the interplay between Hsp90 and DYRK3 in the regulation of SG dynamics, although other Hsp90-dependent/DYRK3-independent mechanisms may also occur.

are also clients of Hsp90 (Taipale *et al*, 2012; Boczek *et al*, 2015). Once stabilized by Hsp90, DYRK3 could autophosphorylate its activation loop to achieve full activation and trigger the disassembly of SGs, by phosphorylating yet unknown molecular targets. In agreement, DYRK3 dynamically shuttles between SGs and the surrounding cytoplasm during the recovery phase after stress (Fig 4C). SG disassembly and reactivation of DYRK3 would then allow translation restoration (Fig 6, upper part). Blocking the loading of DYRK3 onto Hsp90 through Hsp90 inhibition would limit DYRK3 activation, resulting in SG persistence and the progressive destabilization and degradation of DYRK3 (Fig 6, lower part). Future work will be necessary to study in detail how Hsp90 regulates DYRK3 activity and to identify other Hsp90-dependent players that participate, directly or indirectly, in the regulation of SG dynamics.

But why would DYRK3 partition inside SGs during stress in an inactive state when its activity is necessary to initiate SG disassembly and requires binding to Hsp90 in the surrounding cytoplasm? The dynamic partitioning of DYRK3 could have several advantages. First, partitioning into SGs appears to protect DYRK3 from aggregation during stress (Figs 3 and EV3). In fact, targeting aggregation-prone proteins to condensates upon stress is emerging as a general mechanism to prevent irreversible protein aggregation and has lately been described to occur in nucleoli and PML bodies (Frottin *et al*, 2019; Mediani *et al*, 2019). Second, the physical segregation of specific clients of Hsp90 into stress-inducible condensates could decrease the load of clients that require constant assistance by Hsp90 during stress. This could favor the association of Hsp90 with other demanding clients, such as, e.g., folding intermediates that unlike DYRK3 cannot phase separate and would otherwise aggregate during stress. Third, an advantage resulting from the dynamic shuttling of DYRK3 is limiting the amount of active DYRK3 outside of SGs during stress. Clients such as PRAS40 are phosphorylated and inactivated by DYRK3 outside of SGs (Wippich *et al*, 2013), an important step that is required for reactivation of the mTORC1 complex and translation restoration. One possible explanation is that, as long as stress conditions persist, Hsp90 will be occupied with its stress-specific function and will not be able to reactivate DYRK3 and Akt. When stress levels decline, Hsp90 can resume its physiological functions, which include kinase activation and restoration of translation via DYRK3 and Akt.

One important remaining question is how Hsp90 affects the conformation and activity of DYRK3. The fact that under the experimental conditions tested we did not observe Hsp90 colocalization with DYRK3 inside condensates, including SGs and SC35 splicing speckles, suggests that Hsp90 binding to DYRK3 may antagonize its phase separation. Considering that the N-terminal part of DYRK3 is required for its targeting to condensates, our data suggest that the N-terminus of DYRK3 may not be available for phase separation when DYRK3 is bound to Hsp90 (Wippich *et al*, 2013). The finding that DYRK3-dN aggregates upon Hsp90 inhibition suggests that DYRK3 adopts an aggregation-prone, metastable state in the absence of Hsp90 and in the absence of phase separation. This metastable conformation appears to be protected by the presence of the N-terminal segment and the partitioning of DYRK3 into condensates; in line replacement of the N-terminus of DYRK3 with the Sup35-NM domain rescues its targeting to SGs and prevents its aggregation

upon Hsp90 inhibition (Fig EV3). Together these data support the idea that chaperones and solubilizing disordered domains synergize to regulate the aggregation-prone nature of proteins, as suggested previously (Franzmann *et al*, 2018; Franzmann & Alberti, 2019). We propose that Hsp90 could shift DYRK3 conformation toward the active state through cycles of binding and release, similar to what has been previously found for oncogenic v-Src, the most stringent client kinase of Hsp90 (Boczek *et al*, 2015). While interruption of the binding to Hsp90 redirects known client kinases to aggregation or degradation (Taipale *et al*, 2012), it redirects inactive DYRK3 to a novel alternative destination, in a similar manner as direct inhibition of DYRK3 with GSK (Figs 3 and EV3): storage in condensates, where DYRK3 is protected from aggregation and degradation. However, since DYRK3 displays dynamic cycles of partitioning into condensates and the condensed pool is in equilibrium with the surrounding non-condensed pool, prolonged inhibition of Hsp90 will lead to DYRK3 progressive destabilization outside of condensates and degradation (Figs 2, EV2D, and 4D).

The findings that, after dissipation of the stress, ALS patient fibroblasts show a defective induction of DYRK3 expression, together with the reduced expression levels of DYRK3 observed in α-MNs in the lumbar spinal cord of ALS patients bearing the R521C mutation suggest that the deregulated functions of Hsp90 and DYRK3 may contribute to the altered cell stress response and the accumulation of SGs observed in ALS. Globally, these results may provide an additional explanation to why motor neurons are more vulnerable to GA compared to other cell types (Strayer *et al*, 2019). We do not fully understand why the remaining α-MN in FUS-ALS cases that bear large aggregates show reduced immunoreactivity of DYRK3 both in the nucleus as well as in the cytoplasm, nor why the fibroblasts from ALS patients analyzed do not properly upregulate DYRK3 mRNA during the stress recovery phase. This could result from impaired transcription regulation, because of a defective stress response, or from impaired translation restoration, due to defective SG disassembly. We favor the second hypothesis since mRNA degradation is linked to a dynamic equilibrium between polysomes, SGs, and PBs (Decker & Parker, 2012). Future studies should investigate the potential therapeutic value for ALS of pharmacological approaches that boost Hsp90 and DYRK3 functionality.

Conversely, Hsp90 is highly expressed in a large variety of cancers, where it plays complex roles that ultimately promote cancer cell proliferation and inhibition of Hsp90 has been shown to interfere with tumorigenesis (Whitesell & Lindquist, 2005; Calderwood & Neckers, 2016; Rodina *et al*, 2016). Considering that cancer cells are characterized by a high protein synthesis rate, part of the efficacy of the Hsp90 inhibitors may depend on their ability to destabilize DYRK3, thereby promoting SG persistence and perturbing translation restoration after stress. In line, several chemotherapeutic agents induce SGs (Fournier *et al*, 2010), whose targeting has been recently suggested to represent a promising therapeutic avenue for the treatment of cancer (Thedieck *et al*, 2013; Gao *et al*, 2019). Thus, combinations of chemotherapeutics that target SGs with Hsp90 and DYRK3 inhibitors could provide enhanced anti-tumor activity, providing a promising new direction for cancer therapy.

# Materials and Methods

## Reagents and Tools table

| Reagent/Resource | Reference or Source | Identifier or Catalog Number |
|---|---|---|
| **Experimental Models** | | |
| HeLa-Kyoto cell line | Poser *et al* (2008) | N/A |
| HeLa-Kyoto G3BP2-GFP bac cell line | Poser *et al* (2008) | N/A |
| HeLa-Kyoto GFP-FUS WT bac cell line | Mateju *et al* (2017) | N/A |
| HeLa-Kyoto GFP-FUS G156E bac cell line | Mateju *et al* (2017) | N/A |
| HeLa-Kyoto mcherry-G3BP1 bac cell line | Poser *et al* (2008) | N/A |
| HEK293T PABPC1-Dendra-2 | This paper | N/A |
| HeLa Flp-In T-REx V5-HSP70 | This paper | N/A |
| **Recombinant DNA** | | |
| eGFP | Clontech Laboratories, Inc. | Contact company |
| pcDNA5/FRT/TO GFP | Hageman and Kampinga (2009) | N/A |
| GFP-DYRK3 | Wippich *et al* (2013) | N/A |
| GFP-DYRK3-ΔNT | Wippich *et al* (2013) | N/A |
| Sup35-NM-GFP-DYRK3-dN | This study | N/A |
| mcherry-HSP90α | Lev *et al* (2008) | N/A |
| mcherry-HSP90β | Picard *et al* (2006) | N/A |
| myc-RAPTOR | Wippich *et al* (2013) | N/A |
| mRFP-DCP1A | Kedersha *et al* (2008) | N/A |
| pcDNA5/FRT-TO-V5-HSP70 | Hageman and Kampinga (2009) | N/A |
| **Antibodies** | | |
| DYRK3 | Aviva-System Biology | ARP30647_P050 |
| Fibrillarin | EnCor Biotechnology Inc. | MCA-38F3 |
| GFP | Clontech | 632375 |
| G3BP | BD bioscience | 611127 |
| HSPA1A (Hsp70/Hsc70) | StressMarq Biosciences | SMC-104A |
| HSPA8 (Hsc70/Hsp73) | StressMarq Biosciences | SMC-151A |
| HSP90 total | StressMarq Biosciences | SMC-149 |
| HSP90 alpha | StressMarq Biosciences | SMC-147A |
| HSP90 beta | StressMarq Biosciences | SMC-107A |
| myc | Santa Cruz Biotechnology | SC-40 |
| Phospho-PRAS40 (Thr246) | Cell Signaling | 13175 |
| Phospho-p70 S6 Kinase (Thr389) | Cell Signaling | 9205 |
| Phospho-ser10-H3 | Activemotif | 39253 |
| Phospho-4E-BP1 (Thr37/46) | Cell Signaling | 2855 |
| p70 S6 Kinase | Cell Signaling | 9202 |
| PRAS40 | Cell Signaling | 2610 |
| Puromycin (Clone 12D10) | Merck | MABE343 |
| SC-35 | Sigma-Aldrich | S4015 |
| TIA-1 | Santa Cruz Biotechnology | SC-1751 |
| TUBA4A | Sigma-Aldrich | T6074 |
| Ubiquitin | Dako | Z0458 |

**Reagents and Tools table** (continued)

| Reagent/Resource | Reference or Source | Identifier or Catalog Number |
|---|---|---|
| 4E-BP1 | Cell Signaling | 9644 |
| Alexa Fluor™ 594 Azide | Thermo Scientific | A-10270 |
| Donkey anti-Mouse IgG (H + L), Alexa Fluor® 594 | Thermo Scientific | A-21203 |
| Donkey anti-Mouse IgG (H + L), Alexa Fluor® 488 | Thermo Scientific | A-21202 |
| Donkey anti-Mouse IgG (H + L), Alexa Fluor® 647 | Thermo Scientific | A-31571 |
| Donkey anti-Rabbit IgG (H + L), Alexa Fluor® 488 | Thermo Scientific | A-21206 |
| MOUSE IGG HRP LINKED WHOLE AB | GE Healthcare | NXA931 |
| RABBIT IGG HRP LINKED WHOLE AB | GE Healthcare | NA934 |
| **Oligonucleotides and other sequence-based reagents** | | |
| ON-TARGETplus Non-targeting siRNA #1 | Dharmacon | Contact company |
| ON-TARGETplus human HSP90α siRNA | Dharmacon | Contact company |
| ON-TARGETplus human HSP90β siRNA | Dharmacon | Contact company |
| **Chemicals, Enzymes and other reagents** | | |
| Ammonium Chloride | Sigma-Aldrich | 254134 |
| Blasticidin S HCl | Gibco | 12172530 |
| Complete-EDTA | Roche | 11873580001 |
| DAPI | Santa Cruz Biotechnology | SC3598 |
| Duolink™ In Situ Red Starter Kit Mouse/Rabbit | Sigma-Aldrich | DUO92101 |
| Geldanamycin | Enzo Life Science | BML-EI280 |
| GFP-Trap® Agarose beads | Chromotek | Gta-20 |
| GSK626616 | BioVision | B2452 |
| G418-Geneticin | Aurogene | L0015 |
| Hygromycin B | Sigma-Aldrich | H0654 |
| Ly294002 | Sigma-Aldrich | L9908 |
| O-Propargyl-puromycin (OP-puro) | Jena Bioscience | NU-931-05 |
| Puromycin dihydrochloride | Sigma-Aldrich | P8833 |
| Sodium Arsenite | Carlo Erba | S7400 |
| Sodium Arsenite | Sigma-Aldrich | 106277 |
| Tetracycline hydrochloride | Sigma-Aldrich | T7660 |
| VER-155008 | Sigma-Aldrich | SML0271 |
| Wortmannin | Sigma-Aldrich | W1628 |
| Z-Leu-Leu-Leu-al (MG132) | Sigma-Aldrich | C2211 |
| 17-AAG | Enzo Life Science | BML-EI308 |
| **Software** | | |
| Daniel's XL Toolbox | open-source add-in for Microsoft® Excel® | https://www.xltoolbox.net/ |
| Fiji | NIH | https://fiji.sc/ |
| GraphPad Prism6 software | GraphPad | https://www.graphpad.com |
| ScanR Olympus analysis software | Olympus | https://www.olympus-lifescience.com |
| **Other** | | |

## Methods and Protocols

### Experimental model

HeLa-Kyoto bac cell lines were grown at 37°C and 5% $CO_2$ in DMEM high glucose (4.5 g/l) medium supplemented with 2 mM L-glutamine, 100 U/ml penicillin/streptomycin, and 10% fetal bovine serum. The HeLa-Kyoto bac cell lines were kept under selection in geneticin (G-418, Thermo Fisher, 400 μg/ml). The HeLa-Kyoto G3BP1-mCherry cell lines were kept under selection with blasticidin (5 μg/ml). HeLa Flp-in V5-HSP70 was maintained under selection by treatment with blasticidin (5 μg/ml) and hygromycin (0.1 mg/ml). Cells were treated with tetracycline hydrochloride (1 μg/ml) for 72 h to induce Tet-driven transgene expression. We thank Prof. S. Taylor (University of Manchester, UK) for providing HeLa Flp-In

TRex cells, which were used to produce HeLa Flp-in V5-HSP70 following manufacturer instructions (Thermo Fisher Scientific).

Neurons were derived from iPSCs as previously described (Reinhardt *et al*, 2013). All experiments used iPSC-derived neurons matured for 2 weeks.

Human skin fibroblasts were generated from skin biopsies of three healthy donors, one patient affected by familial ALS carrying the R191Q mutation in the gene coding for VCP (Johnson *et al*, 2010) and one sporadic ALS (sALS) patient. All donors provided written informed consent for the collection of skin biopsies. Human fibroblasts were cultured in DMEM high glucose (4.5 g/l) medium supplemented with 2 mM L-glutamine, 100 U/ml penicillin/strepto-mycin, 100 U/ml antibiotic antimycotic solution stabilized, and 10% fetal bovine serum in a 37°C incubator with 5% $CO_2$.

### Method details

#### Transfection, protein extraction, and immunoblotting

Transfections of cDNAs and siRNAs were performed using Lipofec-tamine 2000 (Life Technologies) following manufacturer instructions. Experiments were performed 24 and 72 h after transfection of cDNAs or siRNA, respectively.

To extract total proteins, cells were lysed in Laemmli sample buffer and homogenized by sonication. Prior to separation by SDS–PAGE, the protein samples were boiled for 3 min at 100°C and reduced with β-mercaptoethanol. Proteins were transferred onto nitrocellulose membranes and analyzed by Western blotting.

#### Immunoprecipitation assay

Twenty-four hours post-transfection, HeLa cell lysates were subjected to immunoprecipitation with GFP-Trap® Agarose beads, following manufacturer instructions. In brief, HeLa cells were lysed in Lysis Buffer (10 mM Tris/Cl pH 7.5, 150 mM NaCl, 0.5 mM EDTA, 0.5% NP-40, complete EDTA) for 30 min at 4°C and centri-fuged at 17,000 × *g* for 10 min at 4°C. Cell lysates were incubated with equilibrated GFP-Trap® beads for 1 h at 4°C. After incubation, the immune complexes were centrifuged at 2,500 × *g* for 5 min at 4°C. Beads were washed three times with Wash Buffer (10 mM Tris/Cl pH 7.5, 150 mM NaCl, 0.5 mM EDTA). Both co-immunopre-cipitated proteins and input fractions were resolved on SDS–PAGE and analyzed by Western blotting.

#### Fractionation of NP-40 soluble and insoluble proteins

Twenty-four hours after transfection, cells were harvested in a buffer containing 1% NP-40 (50 mM Tris–HCl, pH 7.4, 150 mM NaCl, 0.25% deoxycholic acid, 1% NP-40, and 1 mM EDTA), passed through a 26G needle 3 times. Cells were lysed on ice for 10 min and then centrifuged at 10,000 × *g* at 4°C for 10 min. The supernatant was collected as NP-40 soluble fraction, while the pellet (NP-40 insoluble fraction) was resuspended with 2% SDS Laemmli buffer.

Protein fractions were boiled for 3 min at 100°C, reduced with β-mercaptoethanol, and separated by SDS–PAGE, followed by Western blotting.

#### Immunofluorescence on cultured cells, Proximity Ligation Assay, and labeling of nascent peptides with OP-puro

Cells were grown on polylysine-coated glass coverslip. After wash-ing with cold PBS, cells were fixed with 3.7% formaldehyde in PBS for 9 min at room temperature, followed by permeabilization with

ice-cold acetone for 5 min at −20°C. Alternatively, cells were fixed with ice-cold methanol for 10 min at −20°C. PBS containing 3% BSA and 0.1% Triton X-100 was used for blocking and incubation with primary and secondary antibodies.

Proximity Ligation Assay was performed with the Duolink™ In Situ Red Kit, using the indicated antibodies (DYRK3 and HSP90) following manufacturer instructions.

Labeling of newly synthesized proteins was performed by incu-bating the cells with 25 μM O-Propargyl-puromycin (OP-puro) for the indicated time points. OP-puro labeled peptides were detected by click chemistry as previously described (Ganassi *et al*, 2016).

#### RNA extraction and RT–qPCR analysis

Human fibroblasts were either left untreated or treated with sodium arsenite 0.5 mM (Carlo Erba Reagents) for 45 min. Then, cells were collected or allowed to recover in drug-free medium for 4 h (recov-ery control). Total RNA was isolated from human fibroblasts using TRIzol reagent (R2050-1-200, Zymo research) and subsequently treated with RNA clean and concentrator Zymo kit (25-R1017, Zymo research) according to the manufacturer's instructions. 0.25 μg of RNA was reverse transcribed using Maxima First strand cDNA Synthesis Kit with dsDNase (K1672, Thermo Fisher) according to the manufacturer's instructions. PCR amplification was performed using Maxima SYBR Green qPCR Master Mix polymerase (Thermo Fisher). The relative changes in the levels of human HSP90AB1, HSP90AA1, DYRK3, and RPL0 mRNAs, the latter used as house-keeping gene, were determined using CFX96 Touch Thermal cycler (Bio-Rad, Hercules, CA, USA) in combination with SYBR green master mix. The primers used were all purchased from Eurofins-Genomics and are listed below: HSP90AB1 For (TGGCAGTCAAG CACTTTTCTGT); HSP90AB1 Rev (GCCCGACGAGGAATAAATAGC); HSP90AA1 For (ATGGCAGCAAAGAAACAC); HSP90AA1 Rev (GTA TCATCAGCAGTAGGGTCA); DYRK3 For (TCCTTCTGAACCACCTCC AC); DYRK3 Rev (CCTTCATCTCACCTCCATCC); RPL0 For (TTAAA CCCTGCGTGGCAATCC); RPL0 Rev (CCACATTCCCCCGGATATGA). The real-time PCR was performed as follows: one cycle of denatura-tion (95°C for 3 min) and 40 cycles of amplification (95°C for 10 s, 60°C for 30 s). A triplicate of each sample was analyzed. Data were analyzed with Bio-Rad CFX Manager 3.1 (Windows 7.0).

#### Live-cell imaging and Fluorescence recovery after photobleaching (FRAP)

FRAP measurements on HeLa cells transfected GFP-DYRK3 WT and dN were performed on a Leica SP8 confocal microscope equipped with a 405 nm and white light lasers using a 63× oil immersion objec-tive. Bleaching and recovery conditions were as follows: 10 s with a laser intensity of 100% at 405 nm for diffuse cytosolic and nucleoplas-mic GFP-DYRK3-dN in untreated cells and 1 s with a laser intensity of 100% at 405 nm for aggregated GFP-DYRK3-dN in cells treated with GA, followed by 300 s. recovery (300 time points) (Fig 3); 1 s with a laser intensity of 100% at 405 nm for SGs containing GFP-DYRK3, followed by 210 s recovery (210 time points) (Fig 4). Analysis of the recovery curves was carried out using a custom written FIJI/ImageJ routine. The equation used for FRAP analysis is as follows $((I_{bleach} - I_{background})/(I_{bleach}(t0) - I_{background}(to)))/((I_{total} - I_{background})/(I_{total}(t0) - I_{background}(to)))$, where $I_{total}$ is the fluorescence intensity of the entire cellular structure, $I_{bleach}$ repre-sents the fluorescence intensity in the bleach area, and

Ibackground the background of the camera offset. Fluorescent density analysis was performed using FIJI/ImageJ and selecting specific region of interest (ROI). When necessary, image drift correction was applied using StackReg plug-in function of the FIJI software suite prior to FRAP analysis. FRAP curves were averaged to obtain the mean and standard deviation.

### Stress granule induction and analysis in HeLa cells and iPSC-derived neurons

SGs were induced in HeLa-Kyoto cells (referred to as Hela cells) and HeLa-Kyoto BAC cell lines expressing either G3BP2-GFP, mCherry-G3BP1, GFP-FUS-WT, or G156E, as well as in the HEK293T PABPC1-Dendra-2 cell line by exposing the cells to different stressors, as described in the figure legends. For the analysis of SG induction, the cells were fixed immediately after exposure to stress and the % of SG-positive and SG-negative cells was counted. The kinetic of SG disassembly was analyzed by either fixing the cells at different time points after removal or the stressor (as detailed in the figure legends), or by live-imaging, followed by quantification of the % of SG-positive and SG-negative cells. SG persistence is defined as the % of SG-positive cells at a given time point during the stress recovery phase.

The iPSC-derived neurons were treated with 500 μM sodium arsenite (Fluka) for 2 h to induce formation of SGs, which contained FUS-eGFP. Afterward, for the "recovery" samples, arsenite-containing medium was removed and replaced with fresh medium for 6 h. 30 μM of the HSP90 inhibitor 17AAG (Selleckchem) was added in the indicated samples. Afterward, neurons were fixed using 4% PFA for 20 min. Hoechst was used for nuclear staining. Confocal microscopy at 40× magnification was used to acquire images, and FUS-eGFP quantification was carried out using Fiji software. Statistically significant differences were determined using GraphPad Prism6 software.

### Human post-mortem tissue

Human post-mortem spinal cord samples fixed in buffered formalin [$n$ = 5 FUS familial ALS patient, $n$ = 5 age-matched controls] were obtained from the archives of the Department of (Neuro) Pathology, Amsterdam UMC, University of Amsterdam, The Netherlands. Both control and fALS cases were selected from a retrospective searchable neuropathologic database reviewed independently by two neuropathologists (E.A. and D.T.) including cases with consent for post-mortem brain/spinal cord autopsy and use of these autopsy tissue as well as medical records for research purpose. All FUS-ALS patients suffered from clinical signs and symptoms of lower and upper MN disease with the eventual involvement of cortex, brain stem motor nuclei. All patients fulfilled the diagnostic criteria for ALS (Ludolph *et al*, 2015). Controls included in the present study were adult individuals without a history of neurological diseases based on the last clinical evaluation. The post-mortem tissues had been obtained within 6–30 h after death.

### Immunofluorescence of lumbar spinal cord tissue sections

Double immunofluorescence staining was performed as previously described (Dreser *et al*, 2017; Marrone *et al*, 2019; Yamoah *et al*, 2020). In brief, deparaffinized tissue sections were heated in citrate buffer, pH 6 (Dako), for 20 min in a pressure cooker. Sections were then blocked (to avoid non-specific bindings) with ready to use 10% normal goat serum (Life Technologies, MD, USA) for 1 h at room temperature before incubating with primary antibody at 4° C overnight. After washing in TBS-T for 10 min, the sections were incubated with Alexa conjugated secondary antibody (1:500 in PBS) at room temperature for 2 h. Sections were washed in TBS-T (2 × 10 min) and stained for 10 min with 0.1% Sudan Black in 80% ethanol to suppress endogenous lipofuscin auto-fluorescence. Finally, the sections were washed for 5 min in TBS-T and mounted with Vectashield mounting medium (Vector Laboratories) containing DAPI. Images from immunofluorescence-labeled sections were taken with a Zeiss LSM 700 laser scanning confocal microscope using 40× and 63× objectives (Zeiss). Images were acquired by averaging 4 scans per area of interest resulting in an image size of 1,024 × 1,024 pixels. The laser intensity was kept constant for all the samples examined. Captured confocal images were analyzed using Adobe Photoshop CS5 and ZEN (Blue edition) 2009 software.

### Quantification and statistical analysis

All statistical analysis were performed using one-way ANOVA, followed by Bonferroni–Holm *post hoc* test for comparisons between three or more groups or Student's *t*-test for comparisons between two groups using Daniel's XL Toolbox or GraphPad Prism6 software.

### Ethics statement

Fibroblast lines from healthy subjects and ALS patients: Protocols and informed consent were approved by the Institutional Ethics Committee (Comitato Etico Provinciale di Modena, Pr. 299/14, 28/04/2015).

Human post-mortem spinal cord samples: The samples were used in compliance with the Declaration of Helsinki. The studies were approved by the Ethical Committees of the Amsterdam UMC (Academic Medical Center; W11_073).

## Data availability

No data were deposited in a public database.

**Expanded View** for this article is available online.

## Acknowledgements

S.C. acknowledges funding from AriSLA Foundation (Granulopathy and MLOpathy); Cariplo Foundation (Rif. 2014-0703); MIUR (Departments of excellence 2018-2022, E91I18001480001 and PRIN, Exo_ALS). S.C. and S.A. are grateful to EU Joint Programme—Neurodegenerative Disease Research (JPND) project. The project is supported through funding organizations under the aegis of JPND (http://www.neurodegenerationresearch.eu/). This project has received funding from the European Union's Horizon 2020 Research and Innovation Programme under grant agreement No 643417. S.A. acknowledges funding from European Research Council (grant number 725836). S.C. and D.K. are grateful to MAECI and The Israeli Ministry of Science and Technology (Dissolve_ALS). D.K. was supported by the European Research Council under the European Union's Seventh Framework Program (FP/2007-2013)/ERC-StG2013 337713 DarkSide starting grant. J.S. acknowledges funding from the Deutsche Forschungsgemeinschaft (DFG) and the CRTD, which is part of the TUD. J.S. was financed by the DFG Research Center (DFG FZ 111) and Cluster of Excellence (DFG EXC 168), including a seed grant. We acknowledge and thank the CMCB Light Microscopy facility for their assistance. We thank CIGS (University of Modena microscopy facility) for technical support and Dr. TM Franzmann (University of Dresden) for

providing the FRAP analysis script. We thank Dr. Picard and Dr. Pelkmans for kindly providing us vectors coding for mCherry-Hsp90 and GFP-DYRK3 WT and dN. We acknowledge the team that contributed to the establishment of the Dutch ALS Tissue Bank, as well as the team that contributed to the collection of ALS tissue samples (Prof. Dr. D. Troost, Prof. Dr. M. de Visser, Dr. A.J. van der Kooi, Dr. J. Raaphorst) and J. Anink (AMC, Amsterdam) for providing technical support. E.A. is supported by the ALS Stichting (grant "The Dutch ALS Tissue Bank") and the neuropathological work-up at the Institute of Neuropathology is supported by the German Research Foundation (DFG; WE 1406/16-1).

## Author contributions

LM performed the majority of the experiments reported in this study with help of FA, ADC, IB, GL, and TT. LM, ADC, and JV performed live-cell imaging, FRAP, and imaging analysis. CC, OP, and JM provided fibroblast lines from healthy donors and ALS patients. VG performed RNA analysis. VT, MC, and JS performed experiments on iPSCs. TA and DK provided HEK293T PABPC1-Dendra2 cells. EA and DT examined and provided the ALS cases; immunohistochemistry on ALS autopsy spinal cord tissue was performed by PT and analyzed by confocal microscopy by AG. SC conceived the project. SC and SA wrote the paper with help of LM and JB.

## Conflict of interest

The authors declare that they have no conflict of interests. Simon Alberti is a scientific advisor of Dewpoint Therapeutics.

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
