## [Review Process File · EMBO Reports]

Hsp90-mediated regulation of DYRK3 couples stress granule disassembly and growth via mTORC1 signaling

Laura Mediani, Francesco Antoniani, Veronica Galli, Jonathan Vinet, Arianna Carrà, Ilaria Bigi, Vadreenath Tripathy, Tatiana Tiago, Marco Cimino, Giuseppina Leo, Triana Amen, Daniel Kaganovich, Cristina Cereda, Orietta Pansarasa, Jessica Mandrioli, Priyanka Tripathi, Dirk Troost, Eleonora Aronica, Johannes Buchner, Anand Goswami, Jared Sternecker, Simon Alberti, and Serena Carra

DOI: 10.15252/embr.202051740

Corresponding author(s): *Serena Carra (serena.carra@unimore.it)* , *Simon Alberti (simon.alberti@tu-dresden.de)*

Review Timeline:	Submission Date:	17th Sep 20
	Editorial Decision:	28th Sep 20
	Revision Received:	14th Dec 20
	Editorial Decision:	21st Jan 21
	Revision Received:	2nd Feb 21
	Accepted:	19th Feb 21

Editor: *Martina Rembold*

Transaction Report: This manuscript was transferred to EMBO reports following peer review at The EMBO Journal.

Referee #1 Review

Received: 21st Aug 20

Report for Author:

The manuscript "Hsp90-mediated regulation of DYRK3 couples SG disassembly and growth via mTORC1 signaling." by Mediani et al. uses a combination of conventional cell biological methods, genetics and fluorescence microscopy in mammalian cell culture models to decipher the role of the

chaperone Hsp90 in regulating stress granule dynamics. They find that Hsp90 affects the activity of the kinase DYRK3, thereby regulating the disassembly of stress granules and restoration of cell growth. This is a thorough and very well controlled study that addresses important aspects of stress granule biology. Although the work unfortunately does not provide mechanistic insights into how DYRK3 controls stress granule resolution, it convincingly establishes a link between Hsp90, DYRK3 function and the disassembly of stress granules, which is certainly of interest to a broad readership.

There are no substantial shortcomings or major concerns, but quite a number of minor issues that should be addressed.

1. Text (highlights, abstract, and several other locations throughout the manuscript): There is no evidence for a direct involvement of DYRK3 in stress granule dissociation; therefore, the authors should provide convincing evidence that DYRK3 acts as a true "dissolvase", or avoid using this word, since this name is misleading as it implies a direct active role.
2. Text (highlights and discussion page 16): "Hsp90 regulates DYRK3 folding": again, the wording is too strong and should be rewritten given that the authors do not provide any data on DYRK3 folding.
3. Fig 1A and all other figures: the authors need to indicate the number of cells, the number independent biological replicates (N) and the statistical tests used for each experiment/panel in the respective figure legends; this important information is often missing. Also, please indicate in the figure legend any abbreviations used in the figure (e.g., "A" should be indicated as "arsenite" in the legend of Fig 3H).
4. Text page 5, line 5: Please state the full name of VCP.
5. Video S6: Ctrl at t=0 has almost no SGs which does not line up with the corresponding quantification (Fig 1A) or video S1. Do you have a more representative ctrl?
6. Fig 1B and S1D + text (highlights, abstract, and page 5): "Our data show that Hsp90 is essential for SG dissolution". This statement is somewhat exaggerated. Neither GA nor 17-AAG nor the KD completely abolishes SG dissolution in both cell lines. Therefore a less strong wording would be more appropriate. Furthermore, the SG persistence kinetics between HeLa and HEK cells do not match well (and the effect of GA is quite different). Do the authors have an explanation for this?
7. Fig S1A and C: please add a label to the figure to indicate more clearly that two different cell lines were used.
8. Fig S1F and corresponding text: "The moderate DRiP accumulation inside SGs did induce changes in SG disassembly kinetics (S1F)" Does this figure really reflect kinetics?
9. Fig 2 and corresponding supplemental figure: It would be helpful to include a GFP control for the IP experiments instead of only an empty vector control.
10. Fig 2 C: x-axis label is missing (also for other PLA quantifications). Furthermore, the authors need to point out in the figure legend that "control" means the normalization to the single antibody control. Also, it is unclear how the normalization was done. Please specify.
11. Fig 2G and H: it is confusing to switch between cells "without SG" and cells "with SG" when generating the graphs. Please be more consistent.
12. Fig 3A and S3G: The results state that cells were treated with GA for 8h but there is no DYRK3 depletion visible. GFP:DYRK3 fluorescence appears even higher in GA treated cells than ctrl cells; this does not fit to the other data (e.g., Fig 2D) or the statements in the text; do the authors have an explanation for this discrepancy? An anti-DYRK3 antibody staining ctrl would be helpful to show that the signal is indeed disappearing upon Hsp90 KD or inhibition.
13. Fig 4A: it is hard to judge from the figure whether DYRK3 and Hsp90 co-localize or not. A quantification with appropriate imaging tools would be very helpful since the signal for Hsp90 is very weak in comparison to the signal of DYRK3 & G3BP1.
14. Fig 4B + text (page 9, bottom): "In normally growing cells, GFP-DYRK3 forms condensates": the

authors should add that this is not always the case, but dependent on the expression level (see Wippich et. al 2013 Fig 2B). Also, the labeling of Fig 4B is confusing. Both, upper and lower panels are labeled differently (HSP90" and "mCherry-HSP90") suggesting the upper one being an immunofluorescence staining. Is this correct or are both images from cells transfected with mCherry-Hsp90 α and β as stated in the figure legend?

15. Fig S4B: "PLA foci indicative of Hsp90/ DYRK3 interaction was generally localized outside DYRK3 condensates": The zoomed in image does not show a condensate, it might make more sense to chose an image with a DYRK3 condensate?

16. Fig 6A-D: There have been several reports that state that WT FUS does not partition into arsenide induced SGs (Bosco et al 2010; Sama et al 2013). Therefore the authors cannot name the WT FUS foci SGs without showing that WT FUS is indeed colocalizing wiht stress granules. The authors need to perform IF staining for SG marker proteins or change axis labeling and rewrite the text accordingly.

17. Fig 6: It would be interesting to test the effect of directly inhibiting DYRK3 by GSK on ALS related cell lines. This would strengthen their conclusion that "these data indicate that impaired SG disassembly because of Hsp90 inhibition specifically affects cells expressing ALS-linked protein variants, suggesting that DYRK3-mediated disassembly may have an important role in ALS pathogenesis".

18. Fig 7: the model lacks the enrichment of DYRK3 into speckles under Hsp90 inhibition without stress (in the lower left panel).

19. The exact information about the experimental procedure regarding SG induction and analysis in HeLa and HEK cells is missing in the materials and methods section. How do the authors define SG persistence?

20. Text (discussion page 19) "We show that Hsp90 does not bind to the N- terminal part of DYRK3, which is required for its targeting to condensates (Wippich et al., 2013).": The authors only show that it can still bind to the truncated form. If the authors want to claim this, they need to provide evidence for that by expressing the GFP-DYRK3-NT construct (Wippich et. al 2013) and perform an IP.

Referee #2 Review

Received: 25th Aug 20

Report for Author:

In their manuscript, Mediani et al describe a role for Hsp90 in the regulation of stress granule disassembly, which is important for cells to resume growth after stress. They find that the stress granule dissolvase DYRK3 associates with HSP90, which occurs outside of stress granules. Functional Hsp90 is required for DYRK stability. Hsp90 inhibitors that disrupt the interaction with DYRK, also impair stress granule disassembly. The authors find that inhibition of Hsp90 either leads to accumulation of DYRK in stress granules, which destabilizes DYRK and targets it for proteasomal degradation. When Hsp90 is active, it promotes stress granule disassembly and restores mTORC1 signaling and growth. The authors show that ALS cells, in which altered stress granules dynamics is part of the pathology, are sensitive to Hsp90 inhibition and that this is associated with a lower induction of DYRK expression. The authors conclude that Hsp90 links stress adaptation to cell viability and growth by regulating the function of the dissolvase DYRK.

The conclusions of the manuscript are overall supported by solid data, some minor issues need clarification.

1. The link between functional Hsp90 and stress granule disassembly is clearly demonstrated. Whether this depends on the interaction between Hsp90 and DYRK is less obvious. As described by the authors, Hsp90 regulates the stability and function of many other signaling molecules. The conclusions of the manuscript are mostly based on inhibition or depletion of Hsp90. Because this will inhibit all other interactions of Hsp90 as well, indirect effects on stress granule dynamics, via other cellular functions, have not been formally excluded. In order to exclude those, the interaction between DYRK and Hsp90 should be exclusively disrupted, for example by a point mutant of DYRK that can no longer interact with Hsp90. Can the authors comment on this?

2. The authors propose that DYRK enters condensates to prevent its irreversible aggregation, which was based on the behavior of mutant DYRK that does not enter condensates but instead forms aggregates. Did the authors test whether the mutant was, except for entering condensates, otherwise functional? Can the authors exclude a role for DYRK in the assemblies that is required for the disassembly, independent of a relocation outside the assembly during recovery from stress and an interaction with Hsp90?

Referee #3 Review

Received: 29th Aug 20

Report for Author:

Stress granules (SG) are biomolecular condensates formed in cytosol upon various stresses. There are only limited data on the mechanism of formation, maintenance and dissolution of SG. Given the significance of SG in degenerative disorders such as ALS, mechanistic insights into SG dynamics is an important area of investigation. In this manuscript, Mediani et al shed light on the role of the molecular chaperone HSP90 in SG dynamics, especially on SG dissolution during recovery. They show that HSP90 inhibition or knock-down results in impairment of SG dissolution, independently of P bodies. The authors propose a mechanism for this finding: the well-known regulator of SG dissolution, the kinase DYRK3, is an HSP90 client. Upon HSP90 inhibition, DYRK3 has reduced abundance, limiting the ability of cells to dissolve SGs during recovery when cells are treated with HSP90 inhibitor. Importantly, the authors show that HSP90-DYRK3-mediated dissolution of SGs is crucial for restoration of translation via mTOR signaling during recovery. Finally, the authors provide some evidence that HSP90-DYRK3 axis may be relevant in ALS pathogenesis. Thus, the study links HSP90 to SG and mTOR signaling during recovery from stress via DYRK3. While the study addresses an interesting question on SG dynamics, it lacks in novelty and mechanistic depth as detailed below.

(1) The role of DYRK3 in SG dynamics and mTOR signaling is already established (PMID: 23415227). Thus the only unknown added by this study is a direct demonstration of DYRK3 as a client of HSP90. However, it is well established that kinases in general are HSP90 clients. There several reports which show that DYRK family of kinases is no exception to this (PMID: 28743892, 26234946). Indeed, DYRK3 itself has already been shown to interact with HSP90 (PMID: 29973724). Thus the study only formally proves that DYRK3 is a client of HSP90, which by itself is neither unexpected nor broadly interesting.

(2) Even if one argues that HSP90-DYRK3 link is new in the context of SG, the mechanistic depth required for this association is completely lacking. For example, the authors have presented multiple lines of evidence that HSP90 regulates SG dynamics, yet it is still not clear whether HSP90 does so exclusively/ majorly through DYRK3. There could be several other mechanisms considering that HSP90 is essential for a large fraction of cellular proteome. DYRK3 as a mechanism for HSP90's role in SG dynamics is only descriptive. One appropriate experiment would be to rescue the defect of

SG dynamics due to HSP90 inhibition by overexpressing the client kinase. This will only be the first step in the right direction, and will likely need to be followed up by mutation analyses of the kinase and the chaperone.

(3) The authors argue that HSP90 is required for DYRK3 activity during SG dynamics; however, the only assay employed for DYRK3 activity is SG dissolution. This is a chicken-and-egg problem: if HSP90 affects SG dynamics via multiple mechanisms (perhaps in addition to DYRK3), then it is not clear if HSP90 is required for DYRK3 activity per se. The change in localization of DYRK3 upon stress and HSP90 inhibition are interesting, but no mechanisms such as protein modifications etc are shown. DYRK3 is likely to autophosphorylate itself when active, giving the authors an opportunity to dig deeper in the question of localization, activity and HSP90-dependence of DYRK3.

(4) The link to ALS is superficial and descriptive, and it is again not clear if DYRK3 is involved at all in the role of HSP90 in SG dynamics in the ALS context (Fig 6A-E). The use of patient fibroblasts is rather tangential as the authors talk about transcriptional regulation of DYRK3 (Fig. 6F, G), which is not the focus of the rest of the manuscript.

(5) The role of HSP90 in translational recovery after stress (Fig. 5) again presents the same caveat - what is the evidence that DYRK3 is involved in this process, other than the circumstantial link with mTOR? Many components of mTOR pathway are HSP90 clients, so there could be several reasons for the observed lack of translational recovery in HSP90-inhibited cells.

(6) Finally, the model (Fig. 7) looks imaginary, lacking evidence for most of the events indicated in the scheme. For example: HSP90 is shown to help DYRK3 just outside the SG. What is the evidence that it is not soluble cytosolic HSP90 doing the job of DYRK3 chaperoning during SG recovery? By mere demonstration of HSP90's presence outside the SG does not implicate this pool of HSP90 in the activation of DYRK3. Also, what is the evidence that SG targeting protects DYRK3 from irreversible aggregation? Fig. 3H used for this interpretation employs N-term deletion of DYRK3. While this mutant may not get into SG, it may also lack additional interactors and hence gets into irreversible aggregation.

Besides these really major issues with this manuscript, there are some minor points that the authors might want to consider:

- Inconsistency in time of recovery (Fig. 5B vs 5C/D), stress paradigm, depiction of data (% cells with SG in Fig. 1E/F vs % cells without SG in Fig. 2G/H).
 - the 1hr timepoint in S11 seems to be missing
 - Change in phosphor state in PRAS40 in 5d
 - The data showing PI3K/akt pathway doesn't have any role in SG dynamics in Fig S5, by inhibition experiments has only videos, doesn't mention the time for which Akt inhibition failed to have any effect.
 - There are no Phospho-p70 S6K and total PRS40 in western blots.
 - In Fig5C, the 4E-BP and p-4E-BP blots look too similar - blot stripping issues?
 - In fig6 Authors DYRK3 show RNA levels but don't show the protein levels or activity which are more relevant
 - In Fig 4B, tagged and endogenous HSP90 seem to behave differently.
 - Fig. 3A-D: wrong labeling in the legend.
 - Very long discussion!
-

Dear Serena

Thank you for the transfer of your research manuscript together with the referee reports from The EMBO Journal to EMBO reports.

We note that the referees, who had evaluated your study for The EMBO Journal, considered your data on the role of Hsp90 in stress granules of potential interest. We agree with referee 3 who pointed out that the role of DYRK3 in this process is already known and that the mechanistic link between Hsp90 and DYRK3 has not been explored in greater depth. However, given the potential interest of Hsp90's role in stress granule disassembly and the fact that EMBO reports has a focus on research papers that report single, key findings with physiological relevance with less emphasis on a detailed mechanistic understanding, we would like to offer you to revise your study for potential publication in EMBO reports, as discussed.

It is not necessary to provide mechanistic insight into how Hsp90 controls DYRK3 localization and activity but the link between the Hsp90-DYRK3 interaction and SG disassembly needs to be strengthened, either by interfering with the interaction or by rescue experiments as suggested by referee 2 (point 1) and referee 3 (point2). Please also provide further data whether DYRK3 inhibition has an effect on ALS-related cell lines (referee 1, referee 3). All other concerns from the referees should be addressed either experimentally or by textual changes and the conclusions should be carefully phrased or toned down.

Given the constructive comments from the referees and the support from at least two referees, we would like to invite you to revise your manuscript with the understanding that the referee concerns (as detailed above and in their reports) must be fully addressed and their suggestions taken on board. Please address all referee concerns in a complete point-by-point response. Acceptance of the manuscript will depend on a positive outcome of a second round of review. It is EMBO reports policy to allow a single round of revision only and acceptance or rejection of the manuscript will therefore depend on the completeness of your responses included in the next, final version of the manuscript.

We invite you to submit your manuscript within three months of a request for revision. This would be December 28th in your case. However, we are aware of the fact that many laboratories are not fully functional due to COVID-19 related shutdowns and we have therefore extended the revision time for all research manuscripts under our scooping protection to allow for the extra time required to address essential experimental issues. Please contact us to discuss the time needed and the revisions further.

- 1) A data availability section is missing.
- 2) Your manuscript contains error bars based on $n=2$. Please use scatter blots showing the individual datapoints in these cases. The use of statistical tests needs to be justified.

Please note that for all articles published beginning 1 July 2020, the EMBO Reports reference style will change to the Harvard style for all article types. Details and examples are provided at <https://www.embopress.org/page/journal/14693178/authorguide#referencesformat>

2) individual production quality figure files as .eps, .tif, .jpg (one file per figure).

Please download our Figure Preparation Guidelines (figure preparation pdf) from our Author Guidelines pages

<https://www.embopress.org/page/journal/14693178/authorguide> for more info on how to prepare your figures.

4) a complete author checklist, which you can download from our author guidelines (<<https://www.embopress.org/page/journal/14693178/authorguide>>). Please insert information in the checklist that is also reflected in the manuscript. The completed author checklist will also be part of the RPF.

5) Please note that all corresponding authors are required to supply an ORCID ID for their name upon submission of a revised manuscript (<<https://orcid.org/>>). Please find instructions on how to link your ORCID ID to your account in our manuscript tracking system in our Author guidelines (<<https://www.embopress.org/page/journal/14693178/authorguide#authorshipguidelines>>)

6) We replaced Supplementary Information with Expanded View (EV) Figures and Tables that are collapsible/expandable online. A maximum of 5 EV Figures can be typeset. EV Figures should be cited as 'Figure EV1, Figure EV2" etc... in the text and their respective legends should be included in the main text after the legends of regular figures.

<<https://www.embopress.org/page/journal/14693178/authorguide#expandedview>>

7) Please note that a Data Availability section at the end of Materials and Methods is now mandatory. In case you have no data that requires deposition in a public database, please state so instead of refereeing to the database.

See also < <https://www.embopress.org/page/journal/14693178/authorguide#dataavailability>>).

Please note that the Data Availability Section is restricted to new primary data that are part of this study.

8) We would also encourage you to include the source data for figure panels that show essential data. Numerical data should be provided as individual .xls or .csv files (including a tab describing the data). For blots or microscopy, uncropped images should be submitted (using a zip archive if multiple images need to be supplied for one panel). Additional information on source data and instruction on how to label the files are available <<https://www.embopress.org/page/journal/14693178/authorguide#sourcedata>>.

9) Our journal encourages inclusion of *data citations in the reference list* to directly cite datasets that were re-used and obtained from public databases. Data citations in the article text are distinct from normal bibliographical citations and should directly link to the database records from which the data can be accessed. In the main text, data citations are formatted as follows: "Data ref: Smith et al, 2001" or "Data ref: NCBI Sequence Read Archive PRJNA342805, 2017". In the Reference list, data citations must be labeled with "[DATASET]". A data reference must provide the database name, accession number/identifiers and a resolvable link to the landing page from which the data can be accessed at the end of the reference. Further instructions are available at <<https://www.embopress.org/page/journal/14693178/authorguide#referencesformat>>.

10) Regarding data quantification:

- Please ensure to specify the name of the statistical test used to generate error bars and P values, the number (n) of independent experiments (please specify technical or biological replicates) underlying each data point and the test used to calculate p-values in each figure legend. Discussion of statistical methodology can be reported in the materials and methods section, but figure legends should contain a basic description of n, P and the test applied.
- Graphs must include a description of the bars and the error bars (s.d., s.e.m.).
- Please also include scale bars in all microscopy images.

11) As part of the EMBO publication's Transparent Editorial Process, EMBO reports publishes online a Review Process File to accompany accepted manuscripts. This File will be published in conjunction with your paper and will include the referee reports, your point-by-point response and all pertinent correspondence relating to the manuscript.

I look forward to seeing a revised version of your manuscript when it is ready. Please let me know if you have questions or comments regarding the revision.

Kind regards,
Martina

Martina Rembold, PhD
Editor

POINT-BY-POINT REPLY TO REFEREES

Referee #1:

The manuscript "Hsp90-mediated regulation of DYRK3 couples SG disassembly and growth via mTORC1 signaling." by Mediani et al. uses a combination of conventional cell biological methods, genetics and fluorescence microscopy in mammalian cell culture models to decipher the role of the chaperone Hsp90 in regulating stress granule dynamics. They find that Hsp90 affects the activity of the kinase DYRK3, thereby regulating the disassembly of stress granules and restoration of cell growth. This is a thorough and very well controlled study that addresses important aspects of stress granule biology. Although the work unfortunately does not provide mechanistic insights into how DYRK3 controls stress granule resolution, it convincingly establishes a link between Hsp90, DYRK3 function and the disassembly of stress granules, which is certainly of interest to a broad readership.

Reply: We appreciate the reviewer positive comments. We are aware that our study lacks the mechanistic insights into how DYRK3 controls SG disassembly. This is particularly difficult since we do not know 1) the molecular target that is phosphorylated by DYRK3 and triggers SG disassembly, and 2) how DYRK3 re-acquires its kinase activity through cycles of partitioning between SGs and the cytoplasm. These aspects need to be studied in detail in the future but will unfortunately require a lot of additional experimental effort that we feel is beyond the scope of this paper.

There are no substantial shortcomings or major concerns, but quite a number of minor issues that should be addressed.

1. Text (highlights, abstract, and several other locations throughout the manuscript): There is no evidence for a direct involvement of DYRK3 in stress granule dissociation; therefore, the authors should provide convincing evidence that DYRK3 acts as a true "dissolvase", or avoid using this word, since this name is misleading as it implies a direct active role.

Reply: The term "dissolvase" was proposed by Rai et al. (2018; PMID: 29973724), who published that DYRK3 is required to disassemble or "dissolve" several types of condensates during mitosis, including stress granules and SC35-splicing speckles. To address this comment, we removed the term "dissolvase" throughout the text as requested and we refer now to SG disassembly.

2. Text (highlights and discussion page 16): "Hsp90 regulates DYRK3 folding": again, the wording is too strong and should be rewritten given that the authors do not provide any data on DYRK3 folding.

Reply: We agree with the referee that our manuscript does not provide direct evidence that Hsp90 regulates DYRK3 folding. However, our work clearly shows that DYRK3 stability is affected upon Hsp90 inhibition (Figure 2). Thus, we changed "DYRK3 folding" to "DYRK3 stability" throughout the text.

3. Fig 1A and all other figures: the authors need to indicate the number of cells, the number independent biological replicates (N) and the statistical tests used for each experiment/panel in the respective figure legends; this important information is often missing. Also, please indicate in the figure legend any abbreviations used in the figure (e.g., "A" should be indicated as "arsenite" in the legend of Fig 3H).

Reply: We added the number of biological replicates, as well as the number of cells quantified where missing. We included the statistical test used in each figure legend. We included the abbreviation "A" for arsenite in the legend of Figure 3H.

4. Text page 5, line 5: Please state the full name of VCP.

Reply: Done as requested.

5. Video S6: Ctrl at t=0 has almost no SGs which does not line up with the corresponding quantification (Fig 1A) or video S1. Do you have a more representative ctrl?

Reply: As requested, we changed the ctrl panel with a more representative one in Video S6 and in Figure EV1J (previous S1L).

6. Fig 1B and S1D + text (highlights, abstract, and page 5): "Our data show that Hsp90 is essential for SG dissolution". This statement is somewhat exaggerated. Neither GA nor 17-AAG nor the KD completely abolishes SG dissolution in both cell lines. Therefore a less strong wording would be more appropriate. Furthermore, the SG persistence kinetics between HeLa and HEK cells do not match well (and the effect of GA is quite different). Do the authors have an explanation for this?

Reply: We rephrased the text accordingly. For example, "Hsp90 is essential for SG dissolution" has been changed with "Hsp90 assists SG disassembly".

Concerning the differential effect of GA in HEK293 cells compared to HeLa cells, we have no other explanation than different cell-type sensitivity to the drug.

7. Fig S1A and C: please add a label to the figure to indicate more clearly that two different cell lines were used.

Reply: We have inserted the name of the cell line used in Fig EV1A, C and D (previous S1A, C and D) as suggested.

8. Fig S1F and corresponding text: "The moderate DRiP accumulation inside SGs did induce changes in SG disassembly kinetics (S1F)" Does this figure really reflect kinetics?

Reply: The referee is correct. We removed this sentence from the revised manuscript.

9. Fig 2 and corresponding supplemental figure: It would be helpful to include a GFP control for the IP experiments instead of only an empty vector control.

Reply: Here we provide for the referee a control experiment showing that in HeLa cells expressing GFP at two increasing concentrations we do not observe interaction with endogenous Hsp90.

10. Fig 2 C: x-axis label is missing (also for other PLA quantifications). Furthermore, the authors need to point out in the figure legend that "control" means the normalization to the single antibody control. Also, it is unclear how the normalization was done. Please specify.

Reply: Thank you for pointing this out. We provided the missing information in the y-axis (average PLA foci/cell) and in the figure legend. Concerning the normalization, in the original version of the manuscript, we normalized to the average number of PLA foci in presence of the DYRK3 antibody alone and the HSP90 antibody alone, which were both used as controls. However, in the revised manuscript, to provide a clearer representation of the results we now calculate all the conditions and negative controls relative to the control condition (which corresponds to untreated cells incubated with both Hsp90 and DYRK3 antibodies; see revised Figure 2C, new panel 2E, and revised Figure EV2B).

For space constraints in revised Figure 2E we did not include a representative image of the PLA foci in untreated versus GSK treated cells (we only show the quantification); the representative images are shown below for the referee.

11. Fig 2G and H: it is confusing to switch between cells "without SG" and cells "with SG" when generating the graphs. Please be more consistent.

Reply: As suggested, we now show in revised Fig 2I (previous 2G) the % of transfected cells with SGs, consistently with panel 2J (previous 2H).

12. Fig 3A and S3G: The results state that cells were treated with GA for 8h but there is no DYRK3 depletion visible. GFP:DYRK3 fluorescence appears even higher in GA treated cells than ctrl cells; this does not fit to the other data (e.g., Fig 2D) or the statements in the text; do the authors have an explanation for this discrepancy? An anti-DYRK3 antibody staining ctrl would be helpful to show that the signal is indeed disappearing upon Hsp90 KD or inhibition.

Reply: Figure 3A shows endogenous DYRK3, which upon short-term treatment with GA is relocalized to nuclear splicing speckles and mitotic bodies. Figure EV3G (before S3G) shows transiently overexpressed GFP-DYRK3 that forms condensates depending on the expression levels (as shown in Wippich et al., and correctly pointed out in comment 14). Upon GA treatment, we observe a general decrease in GFP-DYRK3 levels (data not shown), in agreement with the immunoblotting semiquantitative data shown in Figure 2F, G; however, in cells with higher overexpression levels we can appreciate the relocalization of GFP-DYRK3 to splicing speckles and nucleoli; thus, the images selected in Fig EV3G are qualitative and aim to show this relocalization.

In light of the variability in the expression levels of transiently transfected GFP-DYRK3 from cell to cell, we quantified the relocalization to splicing speckles upon Hsp90 or DYRK3 inhibition only in cells that express endogenous DYRK3 (Fig 3A and EV3A). These results clearly indicate that the recruitment of DYRK3 inside these condensates increases upon Hsp90 or DYRK3 inhibition (in agreement with Figure EV3G). Relocalization to nuclear speckles of overexpressed GFP-DYRK3, but not mCherry-Hsp90, upon treatment with GSK is further shown by a live-cell imaging experiment reported here in reply to referee#2, comment 1 (please see page 7 of this document, third experiment). The other data the reviewer refers to (Figure

2F-H) show the impact of Hsp90 inhibition on the global pool of overexpressed GFP-DYRK3 using total protein extracts and immunoblotting and treatment with GA for 8 to 16 hrs. These data suggest that the pool of exogenous newly synthesized GFP-DYRK3 is very sensitive to Hsp90 inhibition.

13. Fig 4A: it is hard to judge from the figure whether DYRK3 and Hsp90 co-localize or not. A quantification with appropriate imaging tools would be very helpful since the signal for Hsp90 is very weak in comparison to the signal of DYRK3 & G3BP1.

Reply: We now provide in revised Fig 4A the quantification of DYRK3 and HSP90 enrichment inside SGs during the assembly phase. Briefly, SGs were automatically segmented using the G3BP1-mCherry signal and the enrichment of DYRK3 and HSP90 (> 1.5) inside the segmented SGs was calculated using the ScanR (Olympus) software. A total number of 3171 SGs was analyzed. This analysis confirms the lack of colocalization between DYRK3 and Hsp90 inside SGs.

14. Fig 4B + text (page 9, bottom): "In normally growing cells, GFP-DYRK3 forms condensates": the authors should add that this is not always the case, but dependent on the expression level (see Wippich et. al 2013 Fig 2B). Also, the labeling of Fig 4B is confusing. Both, upper and lower panels are labeled differently ("HSP90" and "mCherry-HSP90") suggesting the upper one being an immunofluorescence staining. Is this correct or are both images from cells transfected with mCherry-Hsp90 α and β as stated in the figure legend?

Reply: We now state that condensate formation depends on GFP-DYRK3 expression levels, as correctly pointed out by the referee. Concerning Figure 4B, we specified in the figure and in the figure legend that "HSP90" refers to immunofluorescence staining of endogenous HSP90 (endogenous), while "mCherry-HSP90" refers to transient transfection of mCherry-Hsp90 α and β (exogenous).

15. Fig S4B: "PLA foci indicative of Hsp90/DYRK3 interaction was generally localized outside DYRK3 condensates": The zoomed in image does not show a condensate, it might make more sense to chose an image with a DYRK3 condensate?

Reply: We changed the zoomed image as suggested. Revised Fig EV4B (previous Fig S4B) now shows several GFP-DYRK3 condensates that do not colocalize with PLA foci.

16. Fig 6A-D: There have been several reports that state that WT FUS does not partition into arsenite induced SGs (Bosco et al 2010; Sama et al 2013). Therefore the authors cannot name the WT FUS foci SGs without showing that WT FUS is indeed colocalizing with stress granules. The authors need to perform IF staining for SG marker proteins or change axis labeling and rewrite the text accordingly.

Reply: In agreement with the referee, our data show no recruitment of WT FUS-eGFP inside arsenite-induced SGs in iPSC-MNs. As requested, we now show that the ALS-linked mutant P525L FUS-eGFP is recruited inside sodium arsenite induced SGs by labelling SGs with TIAR (see revised Figure 5B). These results are in line with previous findings published by our collaborator and co-author Dr. Jared Sternecker (Marrone et al., 2018; PMID: 29358088). In this paper, Marrone et al also demonstrated that arsenite-induced P525L FUS-eGFP cytoplasmic foci colocalize with the bonafide SG marker eIF3 in iPSC cells (Figure 2A).

17. Fig 6: It would be interesting to test the effect of directly inhibiting DYRK3 by GSK on ALS related cell lines. This would strengthen their conclusion that "these data indicate that impaired SG disassembly because of Hsp90 inhibition specifically affects cells expressing ALS-linked protein variants, suggesting that DYRK3-mediated disassembly may have an important role in ALS pathogenesis".

Reply: This is a valid point. To address the reviewer's criticism we made the following changes:

- 1) We show that inhibition of DYRK3 significantly delays the disassembly of SGs in iPSC-motor neurons expressing P525L-FUS, linked to ALS (revised Figure 5C, previous Figure 6).
- 2) We include in revised Figure 5F (previous Figure 6) additional data showing that inhibition of DYRK3 with GSK delays the disassembly of SGs in HeLa cells expressing GFP-FUS-WT and the ALS-linked mutant GFP-FUS-G156E.
- 3) We also included in revised Figure 5E (previous Figure 6) additional data showing that depletion of Hsp90 by siRNA delayed SG disassembly in HeLa cells expressing GFP-FUS-WT and G156E, with a slightly stronger impact in the latter ones.
- 4) We stained DYRK3 and FUS in lumbar spinal cord α -motor neurons from healthy subjects and familial ALS patients carrying the R521C mutation in the FUS gene: these new data clearly indicate that surviving α -motor neurons harboring FUS aggregates showed a significant reduction of DYRK3 compared to both α -motor neurons without FUS aggregates in FUS-ALS, and α -motor neurons of the normal controls (Figure 5I and J). Although the reason for this still needs to be worked out, together our results strongly suggests that alterations of Hsp90 and DYRK3 could lead to selective motor neuron vulnerability in ALS.

18. Fig 7: the model lacks the enrichment of DYRK3 into speckles under Hsp90 inhibition without stress (in the lower left panel).

Reply: As suggested, we included in the lower left panel of revised Figure 4D (previous Figure 7) an arrow showing the targeting of DYRK3 into speckles under Hsp90 inhibition without stress.

19. The exact information about the experimental procedure regarding SG induction and analysis in HeLa and HEK cells is missing in the materials and methods section. How do the authors define SG persistence?

Reply: We added the requested technical information in the material and method section (paragraph "Stress granule induction and analysis in HeLa cells and iPSC-derived neurons").

20. Text (discussion page 19) "We show that Hsp90 does not bind to the N- terminal part of DYRK3, which is required for its targeting to condensates (Wippich et al., 2013).": The authors only show that it can still bind to the truncated form. If the authors want to claim this, they need to provide evidence for that by expressing the GFP-DYRK3-NT construct (Wippich et. al 2013) and perform an IP.

Reply: We deleted this sentence from the revised manuscript and we rephrased as follows: "*Considering that the N- terminal part of DYRK3 is required for its targeting to condensates, our data suggest that the N-terminus of DYRK3 is not able to promote targeting to condensates when DYRK3 is bound to Hsp90 (Wippich et al., 2013).*"

Referee #2:

In their manuscript, Mediani et al describe a role for Hsp90 in the regulation of stress granule disassembly, which is important for cells to resume growth after stress. They find that the stress granule dissolvase DYRK3 associates with HSP90, which occurs outside of stress granules. Functional Hsp90 is required for DYRK stability. Hsp90 inhibitors that disrupt the interaction with DYRK, also impair stress granule disassembly. The authors find that inhibition of Hsp90 either leads to accumulation of DYRK in stress granules, which destabilizes DYRK and targets it for proteasomal degradation. When Hsp90 is active, it promotes stress granule disassembly and restores mTORC1 signaling and growth. The authors show that ALS cells, in which altered stress granules dynamics is part of the pathology, are sensitive to Hsp90 inhibition and that this is associated with a lower induction of DYRK expression. The authors conclude that Hsp90 links stress adaptation to cell viability and growth by regulating the function of the dissolvase DYRK.

The conclusions of the manuscript are overall supported by solid data, some minor issues need clarification.

1. The link between functional Hsp90 and stress granule disassembly is clearly demonstrated. Whether this depends on the interaction between Hsp90 and DYRK is less obvious. As described by the authors, Hsp90 regulates the stability and function of many other signaling molecules. The conclusions of the manuscript are mostly based on inhibition or depletion of Hsp90. Because this will inhibit all other interactions of Hsp90 as well, indirect effects on stress granule dynamics, via other cellular functions, have not been formally excluded. In order to exclude those, the interaction between DYRK and Hsp90 should be exclusively disrupted, for example by a point mutant of DYRK that can no longer interact with Hsp90. Can the authors comment on this?

Reply: The referee is right mentioning the possibility that upon HSP90 inhibition or depletion other mechanisms, besides DYRK3 dysfunction, may contribute to delay SG disassembly. To partly address this point, we performed three different experiments.

First experiment:

We verified whether GA delays SG disassembly with a similar efficacy in control versus DYRK3-depleted cells. We find that GA is less efficient in delaying SG disassembly in DYRK3-deficient cells compared to DYRK3-proficient cells (revised Figure EV2G and F). This result supports our interpretation that HSP90 inhibition delays SG disassembly at least in part by impairing DYRK3 stability and function.

Second experiment:

The second experiment is based on the idea that another kinase client of HSP90, casein kinase 2 (CK2), was recently suggested to participate to the regulation of SG dynamics (Reineke et al., 2017; PMID: 27920254). This opens the possibility that, similar to DYRK3, CK2 promotes the disassembly of SGs and that part of the inhibitory effect observed upon GA treatment would be due to the concomitant inhibition of DYRK3 and CK2. We therefore asked whether overexpression of CK2 promotes the disassembly of SGs, similar to what previously reported for overexpression of DYRK3 (Wippich et al., 2013 and our manuscript Figure 2I). Here, we show for the referees that, in contrast to overexpression of DYRK3 (Figure 2I), overexpression of CK2 could not promote the disassembly of SGs during arsenite treatment (see below).

Since we cannot exclude the possibility that, besides DYRK3 destabilization and inhibition, other mechanisms may contribute to regulate SG disassembly, we rephrased the text to make this clear: “Since Hsp90 interacts with 60 % of the human kinome (Taipale et al., 2010), we cannot exclude that other Hsp90 client kinases participate in the process of SG disassembly and translation restoration. Yet, DYRK3-depletion diminished the impact of GA on SG disassembly (Fig EV2G), clearly identifying DYRK3 as one of the Hsp90 targets involved in the regulation of SG dynamics.”

Third experiment:

To address the specific suggestion of the referee that “*the interaction between DYRK and Hsp90 should be exclusively disrupted*”, using two different techniques, pull-down and proximity ligation assay (PLA), we now provide evidence that the treatment of the cells with GSK decreases the interaction between Hsp90 and DYRK3 (see revised Figure 2, new panels D and E). Of note, our interpretation that GSK decreases the interaction between Hsp90 and DYRK3 is further supported by the analysis of GFP-DYRK3 and mCherry-Hsp90 subcellular distribution in living cells exposed to GSK and showing that GFP-DYRK3, but not mCherry-Hsp90, relocalizes into nuclear speckle-like structures (see figure below for the referee).

Together these data demonstrate that: 1) similar to inhibition of Hsp90 with GA, inhibition of DYRK3 with GSK decreases the interaction between Hsp90 and DYRK3; 2) both inhibitors have similar effects on SG dynamics, reinforcing our conclusion that an interplay between Hsp90 and DYRK3 exists and participates to the regulation of SG dynamics.

2. The authors propose that DYRK enters condensates to prevent its irreversible aggregation, which was based on the behavior of mutant DYRK that does not enter condensates but instead forms aggregates. Did the authors test whether the mutant was, except for entering condensates, otherwise functional? Can the authors exclude a role for DYRK in the assemblies that is required for the disassembly, independent of a relocation outside the assembly during recovery from stress and an interaction with Hsp90?

Reply: To address this comment we performed additional experiments.

1) We replaced the N-terminus of DYRK3 with the NM domain of the yeast Sup35 protein, which is known to target proteins into SGs (Gilks et al, 2004). We then studied the targeting of this chimeric protein, referred to as Sup35NM-dN, to SGs and its aggregation propensity upon Hsp90 inhibition. Sup35NM-dN was strongly recruited inside SGs (Fig EV3I). In addition, we show that when GA was added during the recovery phase after arsenite treatment, Sup35NM-dN was sequestered inside persisting SGs, while DYRK3-dN formed perinuclear aggregates (Fig 3I). Next, we also show that, upon treatment of the cells with GA, Sup35NM-dN was diffusely distributed in the cytoplasm (Fig EV3J) and its expression levels progressively decreased (Fig EV3J, K), similar to what observed for DYRK3-WT (Fig 2F). Together these data support the idea that DYRK3 adopts an aggregation-prone metastable state in the absence of Hsp90 and that condensate targeting protects DYRK3 from irreversible aggregation.

2) We experimentally tested whether DYRK3 also plays a role in SG assembly that is required for the disassembly, as suggested by this referee. Briefly, HeLa cells stably expressing mCherry-G3BP1 were treated with sodium arsenite alone or in presence of the DYRK3 inhibitor GSK. Cells were then allowed to recover in drug-free medium; kinetic of SG assembly and disassembly were studied by live-cell imaging. If DYRK3 plays a role during SG assembly that is required for disassembly, DYRK3 inhibition during their formation is expected to delay their disassembly. We found that the kinetics of SG assembly and disassembly were very similar under all condition tested. This result supports the interpretation that DYRK3 activity is specifically required to disassemble SGs. These results are included here for the referee.

A, B: Analysis of the impact of DYRK3 inhibition during arsenite treatment on SG assembly and disassembly kinetics. HeLa kyoto cells stably expressing mCherry-G3BP1 were treated with sodium arsenite (50 μ M) alone or with GSK (5 μ M). A: SG assembly was monitored by live-cell imaging and images were taken over a time period of 45 minutes every 5 min. Dashed lines = 95% confidence intervals. Number of cells counted in three independent experiments: 211 (arsenite); 241 (arsenite + GSK). B: 45 minutes after exposure to sodium arsenite (50 μ M) alone or with GSK (5 μ M), cells were allowed to recover in drug-free medium, and images were taken over a time period of 4 hrs every 10 min. Number of cells counted in three independent experiments: 241 (arsenite, rec.); 382 (arsenite + GSK, rec.). Thus, inhibition of DYRK3 during arsenite treatment does not affect the kinetic of SG assembly nor disassembly compared to cells exposed to sodium arsenite alone.

3) We performed additional FRAP experiments to monitor DYRK3 mobility in the presence of GSK to address the question whether, during the recovery from stress, DYRK3 relocates outside of SGs where it can interact with Hsp90 to be stabilized and regain its activity. Our data show that inactive DYRK3 shuttles between SGs and the surrounding cytoplasm during the recovery phase, similar to what observed in control conditions (Fig 4C, lower panel). However, in presence of GSK, DYRK3 kinase activity is inhibited and DYRK3 cannot promote SG disassembly (Fig EV2E). This result further supports the interpretation that soluble DYRK3 associates with Hsp90 outside of SGs to initiate the disassembly process.

Referee #3:

Stress granules (SG) are biomolecular condensates formed in cytosol upon various stresses. There are only limited data on the mechanism of formation, maintenance and dissolution of SG. Given the significance of SG in degenerative disorders such as ALS, mechanistic insights into SG dynamics is an important area

of investigation. In this manuscript, Mediani et al shed light on the role of the molecular chaperone HSP90 in SG dynamics, especially on SG dissolution during recovery. They show that HSP90 inhibition or knock-down results in impairment of SG dissolution, independently of P bodies. The authors propose a mechanism for this finding: the well-known regulator of SG dissolution, the kinase DYRK3, is an HSP90 client. Upon HSP90 inhibition, DYRK3 has reduced abundance, limiting the ability of cells to dissolve SGs during recovery when cells are treated with HSP90 inhibitor. Importantly, the authors show that HSP90-DYRK3-mediated dissolution of SGs is crucial for restoration of translation via mTOR signaling during recovery. Finally, the authors provide some evidence that HSP90-DYRK3 axis may be relevant in ALS pathogenesis. Thus, the study links HSP90 to SG and mTOR signaling during recovery from stress via DYRK3. While the study addresses an interesting question on SG dynamics, it lacks in novelty and mechanistic depth as detailed below.

(1) The role of DYRK3 in SG dynamics and mTOR signaling is already established (PMID: 23415227). Thus the only unknown added by this study is a direct demonstration of DYRK3 as a client of HSP90. However, it is well established that kinases in general are HSP90 clients. There several reports which show that DYRK family of kinases is no exception to this (PMID: 28743892, 26234946). Indeed, DYRK3 itself has already been shown to interact with HSP90 (PMID: 29973724). Thus the study only formally proves that DYRK3 is a client of HSP90, which by itself is neither unexpected nor broadly interesting.

Reply: The referee is correct in stating that the role of DYRK3 in SG dynamics and mTOR signaling was already established and that DYRK1 was previously shown to interact with Hsp90. However, there are no reports demonstrating a role for Hsp90 in the regulation of SG disassembly, which is the focus of this manuscript. Given the important role of SG in disease and stress response, we think that establishing a link between Hsp90, DYRK3 and the disassembly of SGs is of interest to the growing field of researchers interested in the regulation of condensates in health and disease.

It is also correct that Rai et al. (2018; PMID: 29973724) identified Hsp90 as a chaperone associating with DYRK3 by mass spectrometry (we clearly state this in the text). However, Rai et al did not perform experiments to validate this interaction functionally in vitro or in mammalian cells. Here, we provide a detailed characterization of the interaction between Hsp90 and DYRK3 in mammalian cells and the impact of Hsp90 inhibition on DYRK3 stability and subcellular distribution. All these aspects are novel and pave the way for future studies that will address how Hsp90 mechanistically regulates DYRK3 kinase activity and how this impacts SG dynamics.

(2) Even if one argues that HSP90-DYRK3 link is new in the context of SG, the mechanistic depth required for this association is completely lacking. For example, the authors have presented multiple lines of evidence that HSP90 regulates SG dynamics, yet it is still not clear whether HSP90 does so exclusively/majorly through DYRK3. There could be several other mechanisms considering that HSP90 is essential for a large fraction of cellular proteome. DYRK3 as a mechanism for HSP90's role in SG dynamics is only descriptive. One appropriate experiment would be to rescue the defect of SG dynamics due to HSP90 inhibition by overexpressing the client kinase. This will only be the first step in the right direction, and will likely need to be followed up by mutation analyses of the kinase and the chaperone.

Reply: The referee makes an important point by stating that “it is still not clear whether HSP90 does so exclusively/majorly through DYRK3”. The referee proposed to “rescue the defect of SG dynamics due to HSP90 inhibition by overexpressing the client kinase”. We had also considered to perform this experiment, but we have not been successful for the following reasons. First, overexpression of GFP-DYRK3 reduces the number of cells with SGs during arsenite treatment and this effect was ascribed to its ability to promote SG disassembly (Rai et al., 2018 and our data Fig 2I). We show that GFP-DYRK3 loses this function by addition of GA or 17AAG for 45 min during arsenite treatment (Fig 2I). In addition, inhibition or depletion of Hsp90 destabilizes GFP-DYRK3, which is rapidly degraded by the proteasome (Fig 2F-H). This

makes it technically impossible to rescue the defect of SG dynamics due to Hsp90 inhibition by overexpressing GFP-DYRK3, because in the absence of functional Hsp90 there is no functional DYRK3.

To experimentally address this important point, we performed the following two experiments:

1) we compared the impact of GA on SGs disassembly in control versus DYRK3-depleted cells (Fig EV2F and G). Addition of GA during the recovery phase after sodium arsenite treatment delayed SG disassembly less efficiently in DYRK3-depleted cells compared to control cells (Fig EV2G). Also see reply to Referee #2, comment 1;

2) we tested the potential implication of CK2, a well-known client of Hsp90 that was previously suggested to affect SG dynamics (Reineke et al., 2017; PMID: 27920254). We found that, in contrast to DYRK3, CK2 overexpression could not promote the disassembly of SGs during arsenite treatment (see reply to referee#2, comment number 1). Based on these data, we conclude that Hsp90 regulates SG dynamics, at least in part, by targeting DYRK3 stability and activity.

(3) The authors argue that HSP90 is required for DYRK3 activity during SG dynamics; however, the only assay employed for DYRK3 activity is SG dissolution. This is a chicken-and-egg problem: if HSP90 affects SG dynamics via multiple mechanisms (perhaps in addition to DYRK3), then it is not clear if HSP90 is required for DYRK3 activity per se. The change in localization of DYRK3 upon stress and HSP90 inhibition are interesting, but no mechanisms such as protein modifications etc are shown. DYRK3 is likely to autophosphorylate itself when active, giving the authors an opportunity to dig deeper in the question of localization, activity and HSP90-dependence of DYRK3.

Reply: The reviewer criticizes the lack of data showing how DYRK3 subcellular localization is regulated mechanistically. Although previous studies showed the cycling partitioning of DYRK3 between SGs and the cytoplasm, how this exactly occurs was unknown. In these previous studies, DYRK3 activity was shown to affect not only its ability to promote SG disassembly, but also to directly affect its subcellular localization, as well as to promote the disassembly of SC35 speckles during mitosis (Wippich et al., 2013 and Rai et al., 2018). Based on these studies, we monitored all these three processes (SG disassembly, DYRK3 subcellular localization and accumulation of SC35-positive mitotic bodies) in absence and presence of Hsp90 inhibitors or upon Hsp90 depletion. The changes in DYRK3 subcellular localization that occur upon Hsp90 inhibition mimic those observed when directly inhibiting DYRK3 kinase activity with GSK (Figures 3 and EV3). Next, Hsp90 inhibition leads to an increased number of aberrant SC35-positive mitotic bodies, similar to inhibition of the DYRK3 kinase activity (Figures 3 and EV3). In addition, overexpressed GFP-DYRK3 was shown to reduce the % of cells with SGs during arsenite treatment and this effect was ascribed to its ability to promote SG disassembly during the acute stress; of note this effect requires active DYRK3 (Wippich et al., 2013 and our data, Figure 2I). Here we show that Hsp90 inhibition significantly impaired the ability of overexpressed GFP-DYRK3 to reduce the % of cells with SGs during arsenite treatment (Figure 2I). In line with previous findings (Wippich et al., 2013 and Rai et al., 2018), our data further support the idea that inactive DYRK3 partitions inside condensates and active DYRK3 is required to promote condensate dissociation.

The reviewer asks how DYRK3 autophosphorylation affects its localization and how this is influenced by Hsp90. However, this is a difficult experiment that would require the availability of an antibody that specifically recognizes the phosphorylated form of DYRK3 in the different locations. Our FRAP data show that DYRK3 continuously shuttles from SGs to the surrounding cytoplasm, independently on Hsp90 (Fig 4C, middle panel). Importantly, we now provide evidence that, during the stress recovery phase, DYRK3 dynamically shuttles from SGs to the surrounding cytoplasm also when its kinase activity is inhibited with GSK (Fig 4C, lower panel). Of note, DYRK3 kinase activity is dependent on its autophosphorylation. Thus, together these data suggest that the interaction of DYRK3 with Hsp90 outside of condensates (or at the

border between condensates and their surrounding space) stabilizes DYRK3, enabling its autophosphorylation and activation. This, in turn, would contribute to SG disassembly.

(4) The link to ALS is superficial and descriptive, and it is again not clear if DYRK3 is involved at all in the role of HSP90 in SG dynamics in the ALS context (Fig 6A-E). The use of patient fibroblasts is rather tangential as the authors talk about transcriptional regulation of DYRK3 (Fig. 6F, G), which is not the focus of the rest of the manuscript.

Reply: The reviewer is correct in pointing out that the results provided were preliminary and only suggestive of an implication of altered Hsp90 and DYRK3 in ALS. Altered SG dynamics are an important pathomechanism contributing to ALS. Thus, we think that it is important to publish the implication of Hsp90 and DYRK3 deregulation in altered SG dynamics in ALS cells. To further strengthen this point, which was also requested by Referee #1, comment 17, we provide additional data:

1) we show that inhibition of DYRK3 significantly delays the disassembly of SGs in iPSC-motor neurons expressing P525L-FUS, linked to ALS (revised Figure 5C, previous Figure 6);

2) we now show that depletion of Hsp90 by siRNA delayed SG disassembly in HeLa cells expressing GFP-FUS-WT and G156E, with a slightly stronger impact in the latter ones (revised Figure 5E);

3) we show that inhibition of DYRK3 significantly delays the disassembly of SGs in HeLa cells stably expressing GFP-FUS-WT and the ALS-linked mutant G156E (revised Figure 5F);

4) we stained DYRK3 and FUS in lumbar spinal cord α -motor neurons from healthy subjects and familial ALS patients carrying the R521C mutation in the FUS gene: these new data clearly indicate that surviving α -motor neurons harboring FUS aggregates showed a significant reduction of DYRK3 compared to both α -motor neurons without FUS aggregates in FUS-ALS, and α -motor neurons of the normal controls (Figure 5I). The quantification of the % of α -motor neurons with low or absent DYRK3 and with FUS aggregates is shown (Figure 5J).

(5) The role of HSP90 in translational recovery after stress (Fig. 5) again presents the same caveat - what is the evidence that DYRK3 is involved in this process, other than the circumstantial link with mTOR? Many components of mTOR pathway are HSP90 clients, so there could be several reasons for the observed lack of translational recovery in HSP90-inhibited cells.

Reply: As correctly pointed out by the referee, several components of the mTOR pathway are Hsp90 clients, such as raptor and the Atk/PKB kinase. Moreover, in our paper we confirm that raptor is a client of Hsp90 (Figure EV2C and Delgoffe et al., 2009). We were also concerned about this. This is why we performed a series of experiments to understand whether inhibition of Akt/PKB, which phosphorylates PRAS40 at Thr246 like DYRK3, regulates SG disassembly and the mTORC1 activity (Figure EV5A, B and Videos S8 and S9). These experiments allowed us to determine that direct inhibition of Akt inhibits PRAS40 phosphorylation and impairs translation without affecting SG disassembly kinetics; instead, Hsp90 inhibition, similar to DYRK3 inhibition, impaired SG disassembly kinetics and reactivation of the mTOR pathway. This excludes these proteins as Hsp90 targets in the regulation of SG dynamics, as clearly described in our manuscript: *"Hsp90 promotes translation restoration through the reactivation of two translation-regulatory kinases, DYRK3 and mTORC1, which are gradually released from disassembling SGs. In addition, Hsp90 affects translation restoration through the PI3K/Akt pathway (Giulino-Roth et al, 2017; Ohji et al, 2006), but this pathway is independent of SGs. Thus, cells coordinate translation restoration after stress by activating two synergistic pathways, one SG-dependent and one independent, both of which are highly sensitive to Hsp90 activity (Fig EV5H)."*

(6) Finally, the model (Fig. 7) looks imaginary, lacking evidence for most of the events indicated in the scheme. For example: HSP90 is shown to help DYRK3 just outside the SG. What is the evidence that it is not soluble cytosolic HSP90 doing the job of DYRK3 chaperoning during SG recovery? By mere

demonstration of HSP90's presence outside the SG does not implicate this pool of HSP90 in the activation of DYRK3. Also, what is the evidence that SG targeting protects DYRK3 from irreversible aggregation? Fig. 3H used for this interpretation employs N-term deletion of DYRK3. While this mutant may not get into SG, it may also lack additional interactors and hence gets into irreversible aggregation.

Reply: This is an important point of concern. To exclude this possibility we have gathered the following data: 1) HSP90 is not recruited inside DYRK3-containing condensates (DYRK3 condensates, DYRK3-enriched SC35 nuclear speckles and DYRK3-containing SGs), but it is located outside of these condensates; 2) DYRK3 continuously shuttles between SGs and the surrounding cytoplasm (FRAP data), regardless whether Hsp90 is active or not; 3) DYRK3 continuously shuttles between SGs and the surrounding cytoplasm also in presence of its inhibitor GSK (new FRAP results; Fig. 4C, lower panel). Together these data support our model that DYRK3 associates with Hsp90 outside of SGs to be stabilized and initiate the disassembly process.

Another important point raised by the reviewer was the lack of evidence that SG targeting protects DYRK3 from irreversible aggregation. To address this point, we provide additional data using a chimera that replaces the N-terminus of DYRK3 with the NM domain of the yeast Sup35 protein, which is known to target proteins into SGs (Gilks et al, 2004). The Sup35NM-dN chimera was recruited inside SGs (Fig EV3I). Next, we show that upon addition of GA during the recovery phase after arsenite treatment, Sup35NM-dN was sequestered inside persisting SGs, while DYRK3-dN (which is excluded from SGs) formed perinuclear aggregates (Fig 3I). In addition, we show that upon treatment of the cells with GA, Sup35NM-dN was diffusely distributed in the cytoplasm (Fig EV3J) and its expression levels progressively decreased (Fig EV3J, K), similar to what observed for DYRK3-WT (Fig 2F). Together these data reinforce the idea that condensate targeting protects DYRK3 from irreversible aggregation.

Besides these really major issues with this manuscript, there are some minor points that the authors might want to consider:

- Inconsistency in time of recovery (Fig. 5B vs 5C/D), stress paradigm, depiction of data (% cells with SG in Fig. 1E/F vs % cells without SG in Fig. 2G/H).

Reply: Figure EV5D (previous 5B) shows representative images of cells that have been treated with arsenite, followed by recovery for 2 hrs in drug-free medium (no SGs are left) or in presence of the Hsp90 inhibitors GA or 17AAG, used at two different concentrations. The SGs that persist in presence of the Hsp90 inhibitors contain myc-raptor. In Fig EV5D we did not show pictures of the cells after 4 hrs of recovery time because at this time-point, the only conditions characterized by SG persistency is treatment with 17AAG at the higher concentration (as stated in the text and as shown in Figure 1A, B).

Figure EV5E (previous Fig 5C), instead, shows the phosphorylation of the DYRK3 target protein PRAS40, as well as the mTORC1 target proteins 4E-BP1 and p70 S6K at the later time-point (4 hrs). The choice of the two time-points is not inconsistent, but rather reflects two specific aspects that we want to highlight: 1) sequestration of myc-raptor inside SGs that fail to disassemble upon inhibition of Hsp90 with GA and 17AAG at different concentrations (the 2 hrs time-point here allows us to directly compare all conditions; Figure EV5D) and 2) the phosphorylation status of PRAS40, a target of DYRK3, and p70 S6K and 4E-BP1, targets of mTORC1 kinase, correlates with the extent of SG disassembly since phosphorylation is barely detectable after 4 hrs of recovery in the presence of 5 μ M 17AAG (Figure EV5E).

Figure EV5F (previous 5D) shows that, in absence of other stressors, inhibition of Hsp90 for the longer time used here to study SG dynamics (4 hrs) does not affect the phosphorylation of PRAS40, 4E-BP1 and p70 S6K, in line with the finding that inhibition of Hsp90 does not per se induce the formation of spontaneous SGs (Figure EV1B).

- the 1hr timepoint in S1I seems to be missing

Reply: The 1 hr (60 min) timepoint has been included in revised Fig EV1L (previously named S1L).

- Change in phosphor state in PRAS40 in 5d

Reply: We do not understand this comment (previous Fig 5D is now revised Fig EV5F). In case the comment refers to the slightly lower levels of P-PRAS40 in untreated cells versus cells treated with GA or 17AAG, we would like to point the fact that the TUBA4A levels, used as loading control, are also slightly lower in the control sample.

- The data showing PI3K/akt pathway doesn't have any role in SG dynamics in Fig S5, by inhibition experiments has only videos, doesn't mention the time for which Akt inhibition failed to have any effect.

Reply: In the legend of the Videos referring to the experiments with the Akt inhibitor we reported the time for which Akt inhibition failed to have any effect, which corresponds to 4 hrs (240 min).

"Video S8. Akt inhibition does not induce stress granule formation. Related to Figure EV5.

A time-lapse movie of mCherry-G3BP1 expressing HeLa-Kyoto cells left untreated (CTL) or exposed to the Akt inhibitors Wortmannin (200 nM) or LY294002 (50 μM) for 4 hrs. Pictures were taken every 10 min for up to 240 min."

"Video S9. Akt inhibition does not delay stress granule disassembly. Related to Figure EV5.

A time-lapse movie of mCherry-G3BP1 expressing HeLa-Kyoto cells during the recovery in absence (CTL) or presence of the Akt inhibitors Wortmannin (200 nM) or LY294002 (50 μM) after treatment with sodium arsenite (50 μM) for 45 min. Pictures were taken every 10 min for up to 240 min."

- There are no Phospho-p70 S6K and total PRS40 in western blots.

Reply: We have now repeated these experiments and we prepared a new figure that includes the missing blots as requested (see revised Fig EV5A and B).

- In Fig5C, the 4E-BP and p-4E-BP blots look too similar - blot stripping issues?

Reply: Fig 5C is now revised Fig EV5E: 4E-BP1 and P-4E-BP1 were blotted on two different membranes, therefore excluding any stripping issue. Please, find below the original exposures of the two membranes:

4E-BP1 is phosphorylated at multiple sites and the total antibody used 4E-BP1 (Cell Signaling; 9644) recognizes multiple bands that correspond to unphosphorylated and the differently phosphorylated 4E-BP1 molecules. mTOR can phosphorylate 4E-BP1 at Thr37 and Thr46 and the antibody against P-4E-BP1 used in this study recognizes phosphorylation at both sites (Thr37/46; Cell Signaling, 2855). This explains why the two blots may look somehow similar.

- In fig6 Authors DYRK3 show RNA levels but don't show the protein levels or activity which are more relevant

Reply: Protein analysis by western blotting is less quantitative than mRNA analysis by RT-qPCR and requires higher amounts of cells. ALS primary fibroblasts grow slowly and can be expanded only for a limited number of passages, since they rapidly undergo senescence. Thus, we have access to only limited amount of patient primary fibroblasts. Transcription, like translation, is strongly affected in response to stress and stress-responsive transcription factors play an important role to enable cell adaptation to stress

and regulate their growth capacity (Aprile-Garcia *et al*, 2019; Aramburu *et al*, 2014; Vihervaara *et al*, 2017). Of note, the mTORC1 complex itself can activate stress-responsive transcription factors (Aramburu *et al*, 2014) and Hsp90 itself plays an important role in the regulation of chromatin accessibility by transcription factors (Gvozdenov *et al*, 2019). Based on these reasons, we decided to give priority to the analysis of mRNA levels of DYRK3 and HSP90s in ALS fibroblasts.

However, to address the reviewer concern, we include analysis of DYRK3 protein levels in lumbar spinal cord α -motor neurons from healthy subjects and familial ALS patients carrying the R521C mutation in the FUS gene: these new data clearly indicate a significant reduction of DYRK3 in α -motor neurons with FUS aggregates in FUS-ALS (Figure 5I, J).

- In Fig 4B, tagged and endogenous HSP90 seem to behave differently.

Reply: Endogenous HSP90 shows a rather diffuse cytosolic staining. This is recapitulated by overexpressed mCherry-HSP90 in the majority of the cells. However, as described in our manuscript, under overexpression conditions “we occasionally detected a ring of Hsp90 around” GFP-DYRK3 condensates.

- Fig. 3A-D: wrong labeling in the legend.

Reply: Labeling of Figure 3A and B is correct. Labeling of Figure 3C and D was not clearly explained. We apologize for this and we corrected as follows: (C, D) Quantification of the percentage of mitotic cells showing aberrant SC35 mitotic bodies is reported (C). Number of cells counted: 923 (control), 694 (GA), 951 (17AAG) and 1019 (VER). n = 6 independent experiments, \pm sem, $p < 10^{-10}$. Representative confocal microscopy images of cells with SC35-positive bodies are shown in D.

- Very long discussion!

Reply: We removed from the discussion the paragraph describing the potential targets phosphorylated by DYRK3. We also reorganized the discussion section to make our message clearer.

References

- Aprile-Garcia F, Tomar P, Hummel B, Khavaran A, Sawarkar R (2019) Nascent-protein ubiquitination is required for heat shock-induced gene downregulation in human cells. *Nat Struct Mol Biol* 26: 137-146
- Aramburu J, Ortells MC, Tejedor S, Buxade M, Lopez-Rodriguez C (2014) Transcriptional regulation of the stress response by mTOR. *Sci Signal* 7: re2
- Giulino-Roth L, van Besien HJ, Dalton T, Totonchy JE, Rodina A, Taldone T, Bolaender A, Erdjument-Bromage H, Sadek J, Chadburn A *et al* (2017) Inhibition of Hsp90 Suppresses PI3K/AKT/mTOR Signaling and Has Antitumor Activity in Burkitt Lymphoma. *Mol Cancer Ther* 16: 1779-1790
- Gvozdenov Z, Bendix LD, Kolhe J, Freeman BC (2019) The Hsp90 Molecular Chaperone Regulates the Transcription Factor Network Controlling Chromatin Accessibility. *J Mol Biol* 431: 4993-5003
- Ohji G, Hidayat S, Nakashima A, Tokunaga C, Oshiro N, Yoshino K, Yokono K, Kikkawa U, Yonezawa K (2006) Suppression of the mTOR-raptor signaling pathway by the inhibitor of heat shock protein 90 geldanamycin. *J Biochem* 139: 129-135
- Vihervaara A, Mahat DB, Guertin MJ, Chu T, Danko CG, Lis JT, Sistonen L (2017) Transcriptional response to stress is pre-wired by promoter and enhancer architecture. *Nat Commun* 8: 255

Dear Prof. Carra

Thank you for the submission of your revised manuscript to EMBO reports. We have now received the full set of referee reports that is copied below.

As you will see, all referees acknowledge that the study has been significantly improved during the revision and support publication after some further textual revision. Please discuss your findings with all due caution and avoid any overstatements, in particular on the causality of DYRK3. Other functions of Hsp90 in SG dynamics other than its interaction with DYRK3 should be discussed.

From the editorial side, there are also a few things that we need before we can proceed with the official acceptance of your study:

- Your manuscript contains 5 figures and will be published in our Reports section. This requires that you combine the Results and Discussion section and shorten the main text to 25,000 (+/- 2,000) characters, excluding references and materials and methods. If you feel that an extended discussion is essential, also in light of the referee comments regarding the care that should be taken to avoid overstatements in the results section, please contact me to discuss this further.
- Please reduce the number of keywords to five.
- Please note that all corresponding authors are required to supply an ORCID ID for their name upon submission of a revised manuscript (<<https://orcid.org/>>). This information is still missing for Dr. Alberti. Please find instructions on how to link your ORCID ID to your account in our manuscript tracking system in our Author guidelines (<<https://www.embopress.org/page/journal/14693178/authorguide#authorshipguidelines>>)
- Author Contributions: please specify the contribution of Giuseppina Leo.
- Please add callouts to Movies EV3 and EV4 where appropriate.
- Movies: Please remove the movie legends from the article file and provide them as individual README.txt files. Then ZIP each legend together with its movie and upload the zipped files.
- Please also correct the names of the movie files from Video Sx to Movie EVx.
- Please move the figure legends to the end of the Article file.
- I attach to this email a related manuscript file with comments by our data editors. Please address all comments and upload a revised file with tracked changes with your final manuscript submission.
- During our routine analysis of all images and figure panels before publication, we detected some unexplained features in Figure EV5C. In order to avoid any ambiguities, we kindly ask you to provide the unmodified source data for these figure panels.
- Thank you for sending bullet points highlighting the main findings of the article. Could you please also provide a 1-2 sentence draft that summarizes the key findings of your work? In addition, we need a synopsis image that is 550x200-600 pixels large (width x height) in .png format. You can either show a model or key data in the synopsis image. Please note that the size is rather small and

that text needs to be readable at the final size. Please send us this information along with the revised manuscript.

With kind regards,

Martina Rembold

Referee #1:

The authors have partly addressed my main concerns in the revised version. I would nonetheless suggest to the authors to tone down the claims made on direct causality as well as molecular mechanism, when these are based on circumstantial data. For example, HSP90's ability to chaperone the kinase just outside the SG has not been shown. Authors are advised to make such models in the (already long) discussion of the paper rather than in the results/ abstract/ highlights section. Such conclusions do not have strong experimental basis, but only circumstantial evidence. This is true for several other claims that authors should textually alter by using phrases such as 'suggest', 'may', 'possible explanation' etc in results, and further expand in Discussion when necessary.

Also, I would suggest that the authors do not comment on the 'centrality' of HSP90 in SG dissolution. There must be several proteins which when downregulated cause SG disassembly defect - are they all 'central hubs'?

Referee #2:

The authors have fully addressed all of my concerns. I would only suggest changing the name from Sup35NM-dN to Sup35NM-DYRK3-dN throughout the text, as omitting "DYRK3" could be misleading.

Referee #3:

Just a minor note:

Still, other interactions of Hsp90, those that are not directly involved in the mechanisms of stress granule (dis)assembly, are likely to be important for the health of cells and the way cells respond to and recover from stress. These functions will also be impaired by Hsp90 inhibition and may indirectly influence stress granule dynamics, simply because cells are more unhealthy when HSP90 is inhibited. While more evidence is added by the authors for the contribution of DYRK3, the possibility that functions of Hsp90, other than its interaction with DYRK3, contribute to the observed effects

has not been fully excluded. The authors could include a sentence to acknowledge this possibility in their discussion.

POINT-BY-POINT REPLY TO REFEREES EMBOR-2020-51740V3

Referee #1:

The authors have partly addressed my main concerns in the revised version. I would nonetheless suggest to the authors to tone down the claims made on direct causality as well as molecular mechanism, when these are based on circumstantial data. For example, HSP90's ability to chaperone the kinase just outside the SG has not been shown. Authors are advised to make such models in the (already long) discussion of the paper rather than in the results/ abstract/ highlights section. Such conclusions do not have strong experimental basis, but only circumstantial evidence. This is true for several other claims that authors should textually alter by using phrases such as 'suggest', 'may', 'possible explanation' etc in results, and further expand in Discussion when necessary.

Reply: As suggested we modified our conclusions throughout the text using words such as “suggest”, “would”, “one possible explanation”. We also moved the “Schematic model showing how Hsp90 regulates SG dynamics via DYRK3” from Fig 4D to Fig 6. As correctly pointed out by the referee “HSP90's ability to chaperone the kinase just outside the SG has not been shown”. We now state that “*Hsp90 interacts with non-condensed DYRK3 and regulates its stability and activity outside or at the boundary of condensates*”. Moreover, in the revised manuscript, the model is not described anymore in the results section, but in the discussion section. We also clearly state that “*Future work will be necessary to study in detail how Hsp90 regulates DYRK3 activity and to identify other Hsp90-dependent players that participate, directly or indirectly, in the regulation of SG dynamics.*”

We also revised the highlights to avoid overstatements.

Also, I would suggest that the authors do not comment on the 'centrality' of HSP90 in SG dissolution. There must be several proteins which when downregulated cause SG disassembly defect - are they all 'central hubs'?

Reply: We deleted the words “central hub” and we rephrased the last sentence of the abstract as follows: “*Thus, Hsp90 links stress adaptation and cell growth by regulating the activity of a key kinase involved in condensate disassembly and translation restoration.*”

Referee #2:

The authors have fully addressed all of my concerns. I would only suggest changing the name from Sup35NM-dN to Sup35NM-DYRK3-dN throughout the text, as omitting "DYRK3" could be misleading.

Reply: We thank the referee for acknowledging that we addressed all his/her concerns. We changed the name of Sup35NM-dN to Sup35NM-DYRK3-dN throughout the text as suggested.

Referee #3:

Just a minor note:

Still, other interactions of Hsp90, those that are not directly involved in the mechanisms of stress granule (dis)assembly, are likely to be important for the health of cells and the way cells respond to and recover from stress. These functions will also be impaired by Hsp90 inhibition and may indirectly influence stress granule dynamics, simply because cells are more unhealthy when HSP90 is inhibited. While more evidence is added by the authors for the contribution of DYRK3, the possibility that functions of Hsp90, other than

its interaction with DYRK3, contribute to the observed effects has not been fully excluded. The authors could include a sentence to acknowledge this possibility in their discussion.

Reply: We thank the referee for recognizing that all major concerns were addressed. To address his/her last minor note, we have revised the discussion as follows: *“Since Hsp90 interacts with 60 % of the human kinome (Taipale et al., 2010), we cannot exclude that other Hsp90 client kinases participate in the process of SG disassembly and translation restoration. Yet, DYRK3-depletion diminished the impact of GA on SG disassembly (Fig EV2G), clearly identifying DYRK3 as one of the Hsp90 targets involved in the regulation of SG dynamics. Furthermore, Hsp90 was recently shown to stabilize several SG components (O'Meara et al., 2019). This finding suggests that defective SG disassembly upon Hsp90 inhibition could also result from the destabilization of other yet unknown SG components.”*

Finally, we moved the working model (previous Fig 4D) to Fig 6, and we describe this model in the discussion, concluding as follows: *“Future work will be necessary to study in detail how Hsp90 regulates DYRK3 activity and to identify other Hsp90-dependent players that participate, directly or indirectly, in the regulation of SG dynamics.”*

Prof. Serena Carra
Università degli Studi di Modena e Reggio Emilia
Dipartimento di Scienze Biomediche, Metaboliche e Neuroscienze
Giuseppe Campi 287
Modena, Modena 41125
Italy

Dear Prof. Carra,

I am very pleased to accept your manuscript for publication in the next available issue of EMBO reports. Thank you for your contribution to our journal.

At the end of this email I include important information about how to proceed. Please ensure that you take the time to read the information and complete and return the necessary forms to allow us to publish your manuscript as quickly as possible.

As part of the EMBO publication's Transparent Editorial Process, EMBO reports publishes online a Review Process File to accompany accepted manuscripts. As you are aware, this File will be published in conjunction with your paper and will include the referee reports, your point-by-point response and all pertinent correspondence relating to the manuscript.

If you do NOT want this File to be published, please inform the editorial office within 2 days, if you have not done so already, otherwise the File will be published by default [contact: emboreports@embo.org]. If you do opt out, the Review Process File link will point to the following statement: "No Review Process File is available with this article, as the authors have chosen not to make the review process public in this case."

Should you be planning a Press Release on your article, please get in contact with emboreports@wiley.com as early as possible, in order to coordinate publication and release dates.

Thank you again for your contribution to EMBO reports and congratulations on a successful publication. Please consider us again in the future for your most exciting work.

Yours sincerely,

THINGS TO DO NOW:

You will receive proofs by e-mail approximately 2-3 weeks after all relevant files have been sent to our Production Office; you should return your corrections within 2 days of receiving the proofs.

Please inform us if there is likely to be any difficulty in reaching you at the above address at that time. Failure to meet our deadlines may result in a delay of publication, or publication without your corrections.

All further communications concerning your paper should quote reference number EMBOR-2020-51740V3 and be addressed to emboreports@wiley.com.

Should you be planning a Press Release on your article, please get in contact with emboreports@wiley.com as early as possible, in order to coordinate publication and release dates.

Corresponding Author Name: Serena Carra

Journal Submitted to: EMBO reports

Manuscript Number: EMBOJ-2020-51740V3